



# Constraining 3D geometric gravity inversion with 2D reflection seismic profile using a generalized level-set approach: application to Eastern Yilgarn craton

Mahtab Rashidifard[1, 3], Jérémie Giraud[1, 3], Mark Lindsay[1, 3], Mark Jessell[1, 3], and Vitaliy Ogarko[1, 2]

[1] Centre of Exploration Targeting (School of Earth Sciences), University of Western Australia, 35 Stirling Highway, Crawley, WA, 6009, Australia

[2] International Centre for Radio Astronomy Research (ICRAR), University of Western Australia, 7 Fairway, Crawley, WA 6009, Australia

[3] Mineral Exploration Cooperative Research Centre, School of Earth Sciences, University of Western Australia, 35 Stirling Highway, WA Crawley 6009, Australia

*Correspondence to*: Mahtab Rashidifard (Mahtab.rashidifard@research.uwa.edu.au)

**Abstract.** One of the main tasks in 3D geological modelling is the boundary parametrization of the subsurface from geological observations and geophysical inversions. Several approaches have been developed for geometric inversion and joint inversion of geophysical datasets. However, the robust, quantitative integration of models and datasets with different spatial coverage, resolution, and levels of sparsity remains challenging. One promising approach for recovering the boundary of the geological units is the utilization of a level-set inversion method with potential field data. We focus on constraining 3D geometric gravity inversion with sparse lower-uncertainty information from a 2D seismic section.

We use a level-set approach to recover the geometry of geological bodies using two synthetic examples and data from the geologically complex Yamarna terrane (Yilgarn craton, Western Australia). In this study, a 2D seismic section has been used for constraining the location of rock unit boundaries being solved during the 3D gravity geometric inversion. The proposed work is the first we know of that automates the process of adding spatially distributed constraints to the 3D level-set inversion. In many hard-rock geoscientific investigations, seismic data is sparse and our results indicate that unit boundaries from gravity inversion can be much better constrained with seismic information even though they are sparsely distributed within the model. Thus, we conclude that it has the potential to bring the state of the art a step further towards building a 3D geological model incorporating several sources of information in similar regions of investigation.

## 1 Introduction

Inverted models from geophysical inversions have broad applications in 3D geological modelling if they specify distinct rock units, rather than just petrophysical distributions. One way to achieve this is by using geometric inversion approaches. These methods are receiving increasing attention in geophysical inverse problems with a focus on recovering the shape of different

rock units. Using several geophysical techniques that enable us to recover the geometry of the specified rock type leads to an inverted model consistent with geophysical datasets which is compatible with geological interpretations.

Gravity data is one of the most widely modelled geophysical dataset worldwide. The inversion of gravity datasets can either

be performed with the aim of retrieving density contrasts (Boulanger and Chouteau, 2001; Lamichhane and Gross, 2017; Li and Oldenburg, 1998; Martin et al., 2020; Ogarko et al., 2021) or depth and shapes (geometry) of unit boundaries (Cai & Zhdanov, 2015; Li & Qian, 2016). Geometric inversion approaches generate models with distinct density contrast units suitable for geological modelling purposes (Jessell et al., 2014; Leliévre et al., 2015; Lindsay et al., 2013). As complementary information is required to compensate for poor vertical resolution of gravity datasets (Coutant et al., 2012; Lelièvre et al.,

2010; Sun and Li, 2015), the geometry of the retrieved models from geometric gravity inversion are more plausible if the models are compensated in depth with other geophysical datasets.

On the other hand, seismic images provide higher vertical resolution of deep structures and the detectible horizons from these images can be correlated with the density contrast surfaces. This enables us to utilize seismic information from the subsurface structures to constrain the gravity inversion. Combining these interpretations with a geometric gravity inversion is a step toward

a geologically plausible gravity inversion in agreement with seismic images.

The level-set approach being widely applied in geometry optimization problems (Osher et al., 2004) can be applied to geophysical inversion techniques by parametrizing the rock unit boundaries implicitly as iso-contours of higher dimensional functions ( Li et al., 2016). Boundaries are then optimized during the inversion by evolving these level-set functions (Burger and Osher, 2005). The defined units in the model can then be merged and separated or even omitted during the inversion based

on topological rules (Cai and Zhdanov, 2015;van Zon and Roy Chowdhury, 2010).

Recovering rock unit boundaries using a level-set gravity inversion method has been studied in recent years with the focus on inversion for different numbers and shapes of the buried bodies (Cai & Zhdanov, 2017; Farquharson, Ash, & Miller, 2008; Leliévre et al., 2015; W. Li, Lu, & Qian, 2016; Zheglova et al., 2018). However, automatic geologically conditioned geophysical level-set inversion has not been addressed in the aforementioned studies. This is because the proposed methods

are either limited to a specific number of units (level-set functions) or are bound to a specific type of topology by using explicit modelling for defining rock units.

We utilize a generalized level-set inversion technique introduced by Giraud et al (2021a). This approach extends level-set methods previously developed by Cardiff and Kitanidis, (2009) and Li et al (2017). This generalized level-set algorithm, not

only allows us to define an arbitrary number of density contrast units free of shape limitation but also allows us to add sparsely
distributed low uncertainty data to constrain the gravity inversion. We use information from the seismic section as low
uncertainty data as in large-scale studies borehole data might not necessarily be available for constraining purposes. For more
details about the methodology, we refer readers to (Li et al., 2017; Raponi et al., 2017; Tai & Chan, 2004; Giraud et al., 2021).

In this study, we focus on constraining surface gravity data that possesses good lateral resolution with a 2D reflection seismic
profile that traverses the study area. In this manner, sparsely distributed low-uncertainty data from the seismic section can be
utilized to guide 3D gravity inversion results. The proposed work is the first we know of that quantitatively integrates the
geometries from the seismic section into 3D gravity inversion. Our results suggest that the proposed approach has the potential
to bring the state of the art a step further towards automatically building a 3D geological model which is consistent with
available geological and geophysical datasets.

The remainder of this paper is organized as follows. We first provide a summary of the generalized level-set method we use
and show its applicability to gravity inversion constrained by sparser seismic data and geological knowledge. We then apply
the proposed approach on two 3D synthetic datasets to show the proof-of-concept. Subsequently, we present a case study at
the geologically complex Yamarna terrane in Yilgarn Craton, Western Australia. We use the 3D surface gravity datasets and
a 2D seismic profile available in this area to apply the constrained inversion approach. Finally, the performance of the
constrained level-set approach on the synthetic and field studies and the resulting models are reviewed and assessed in the
discussion section.

## 2 Method

### 2.1 Generalized level-set method

We use the generalized level-set inversion formulation introduced by Giraud et al (2021a) for our constrained inversion
problem. We extend their work to the use of sparse seismic constraints and topological rules. A summary of the method is
provided below. We refer to Giraud et al (2021a) for details about the mechanics of the inverse problem and use the same
notation as these authors in what follows.

First, a geologically plausible starting model is defined with density contrasts $\Delta\boldsymbol{\rho}_{k=1,..,N}$ assigned to each of the N rock units.
Then for each unit, signed-distance values to interfaces ($\boldsymbol{\phi}_k$) are calculated using the fast marching method (Sethian, 1999),
where the outline of a given unit is defined by $\boldsymbol{\phi}_k = 0$. A smeared-out Heaviside function is then defined to transform the
signed-distance functions to a multinary structure. This allows us to generate a physical property model ($\mathbf{m}$, with M number of
cells) from the signed-distance values. During inversion, we restrict the evolution of the model between two successive





iterations to the interface between rock units. It is controlled by a parameter ($\tau$) defining the thickness of the interfaces between units. It can be dependent on the cell size or chosen arbitrarily. $\tau$ is an important parameter controlling the search space in the vicinity of the interface in each model update. The topological stability of the model is then ensured prior to generating the
physical property model by allowing only one positive value of $\boldsymbol{\phi}$ at any location in the model.

Initializing the model space as a function of signed-distance values requires the definition of a new sensitivity matrix measuring the sensitivity (writing its individual elements $\mathbf{J}_{ij}^{\boldsymbol{\phi}_k}$) of each observed gravity data ($\boldsymbol{i}$) to changes of each signed-distance function ($\boldsymbol{\phi}_k$) in each model cell ($j$). This sensitivity matrix is then used in a least-squares inversion formulation as in Eq. (1). In this framework, residuals of the calculated and observed datasets ($\boldsymbol{r} = [\boldsymbol{d}^{obs} - \boldsymbol{d}^{calc}]^{T}$) are minimized during the
inversion. The data misfit term to optimize can be written as follows:

$$\Psi^{r} = \left\| \boldsymbol{J}^{\phi} \delta\boldsymbol{\phi} - (\boldsymbol{d}^{obs} - \boldsymbol{d}^{calc}) \right\|_{2}^{2}, \tag{1}$$

Where $\Psi^{r}$ represents the misfit function.

The misfit function is minimized iteratively to solve for changes in the signed-distance function ($\delta\boldsymbol{\phi}_k$). To stabilize the inversion problem and incorporate prior information, regularization terms are added to Eq. (1) as discussed below. We use an
updated form of the regularizations as spatial constraints to the inversion problem based on seismic information using the aforesaid method. To encourage geological plausibility, we then introduce the utilization of topological rules on the resulting model during inversion. The utilization of seismic information to define regularizations is introduced below.

## 2.2 Regularized level-set inversion

Other level-set inversion approaches (Li et al., 2016, 2017; Zheglova et al., 2018) apply regularizations to the misfit function.
In such cases, the problem is usually regularized by favouring structures with the shortest interface overall length (2D case) or smallest surface (3D case). Smoothing the geometries characterized by the zero level-set functions by minimizing the length, generates shapes with the smallest area and regularizing these inversion problems can be limited to the specific shape of units that can introduce a bias towards unrealistically simple geometries. Using known density contrast values for the model parameterization in these approaches significantly reduces the non-uniqueness of the inversion. In addition, regularizing the
inverted model using prior information may reduce the non-uniqueness of the inverse problem further. In the level-set inversion scheme we use, prior information can be appended as regularization terms ($\Psi^{t}$) in the same fashion as that smallness terms regularize inversion problems (Calvetti et al., 2000). This term is appended to the misfit function as below:

$$\Psi(\delta\boldsymbol{\phi}, r) = \Psi^{r}(\delta\boldsymbol{\phi}, r) + \Psi^{t}(\delta\boldsymbol{\phi}), \tag{2}$$

Where;




$$\Psi^{t}(\delta\boldsymbol{\phi}) = \|\mathbf{W}\delta\boldsymbol{\phi} - \boldsymbol{q}\|_{2}^{2}, \tag{3}$$

$$\mathbf{W} = \begin{bmatrix} \boldsymbol{W}_{S} \\ \mathbf{W}_{P} \end{bmatrix} \tag{4}$$

Where $\Psi^{t}$ tends to minimize the total update in the signed-distance ($\delta\boldsymbol{\phi}$) and acts as a smallness term that encourages the signed-distance update ($\boldsymbol{\delta\phi}$) of rock units to reach specific values stored in $\boldsymbol{q} = (\mathbf{0}, \boldsymbol{v}_{1}, \boldsymbol{v}_{2}, \dots, \boldsymbol{v}_{N})$. Here, $\boldsymbol{W}_{S}$ is a global

regularization $1 \times MN$ vector, and $\mathbf{W}_{P}$ is a local regularization $N \times MN$ matrix that has $\boldsymbol{W}_{p_{k=1,2,\dots N}}$ as rows. Both $\boldsymbol{W}_{S}$ and $\mathbf{W}_{P}$ encapsulate prior information for the inversion.

Inverting for the total changes in the signed-distance value at each model update and controlling these changes through regularizations enables us to extend the approach for the constrained inversion problems. In the next subsection, we introduce the method we developed to incorporate 2D information into the regularization scheme presented here, using the case of 2D

seismic data. We show that updating regularization terms based on low-uncertainty datasets extends the application of global and local regularizations to act as global and local constraints.

**2.3 Translating seismic information to spatial constraints**

Global ($\boldsymbol{W}_{S}$) and local ($\boldsymbol{W}_{p_{k=1,\dots,N}}$) regularization terms are appended to the sensitivity matrix of the level-set method as in Eq. (5) as uniform vectors in a system of equations solved in the least-squares sense. They tend to stabilize changes of signed-

distance functions as damping terms. This supports the capability of regularization terms to perform the constraining process using low-uncertainty datasets such as seismic as weighing terms. We propose that adding unevenly distributed weights to the regularizations based on information from seismic in depth can act as constraints for the level-set inversion. Therefore, as long as the interpreted or inverted sections from the seismic are included in vectors with the same size as the model, any primary information from pre-existing modelling such as seismic sections can be translated to weighting terms to define constraints.

$$\begin{bmatrix} \mathbf{J}^{\boldsymbol{\Phi}_{1}} \ \mathbf{J}^{\boldsymbol{\Phi}_{2}} \ \dots \ \mathbf{J}^{\boldsymbol{\Phi}_{N}} \\ \boldsymbol{W}_{S} \\ \boldsymbol{W}_{p_{1}} \\ \boldsymbol{W}_{p_{2}} \\ . \\ . \\ \boldsymbol{W}_{p_{N}} \end{bmatrix} \begin{bmatrix} \delta\boldsymbol{\phi}_{1} \\ \delta\boldsymbol{\phi}_{2} \\ . \\ . \\ \delta\boldsymbol{\phi}_{N} \end{bmatrix} = \begin{bmatrix} \boldsymbol{r} \\ 0 \\ v_{1} \\ v_{2} \\ . \\ . \\ v_{N} \end{bmatrix} \tag{5}$$

Strategies for transferring the knowledge from a seismic section to a weighting term that can be used as sparse constraints can vary depending on the seismic data and availability of other datasets. Overall, the interpretation from a vertically extended seismic profile can be assumed as an interpreted sample geological model. In Fig. 1, we assume that interpretation has led to





a conceptual geological model comprising of N lithological units. Therefore, 2D seismic interpretations are treated in the same
fashion as a 2D geological surface map to constrain the gravity inversion at depth. Constructing the weighted regularization
terms or constraint terms based on available interpretation is necessary before starting the inversion. All arrays of the model
that lie within the sample section are weighted accordingly for all lithologies as $\boldsymbol{w}_{s_{k=1,\dots,N}}$ (all of dimensions $1 \times M$) which then
construct the regularization terms as in Eq. (6). Thus, the values of $\boldsymbol{W}_S$ and $\boldsymbol{W}_{p_{k=1,\dots,N}}$ are adjusted locally accordingly with
seismic interpretations to favour the preservation of interpreted units along the seismic section. The structure of weighting
vectors from seismic interpretations within regularization terms is as follow:

$$
\begin{cases}
\boldsymbol{W}_S = [\boldsymbol{w}_{s_1}, \boldsymbol{w}_{s_2}, \dots, \boldsymbol{w}_{s_N}] \\
\boldsymbol{W}_{p_1} = [\boldsymbol{w}_{s_1}, \boldsymbol{0}, \dots, \boldsymbol{0}] \\
\boldsymbol{W}_{p_2} = [\boldsymbol{0}, \boldsymbol{w}_{s_2}, \dots, \boldsymbol{0}] \\
\qquad \cdot \\
\qquad \cdot \\
\boldsymbol{W}_{p_N} = [\boldsymbol{0}, \boldsymbol{0}, \dots, \boldsymbol{w}_{s_N}]
\end{cases}
\tag{6}
$$

Where, $\boldsymbol{0}$ is a zero vector of dimensions $1 \times M$.

Figure 1 shows the weighting matrix for a given lithology (Fig. 1a) and its location within regularization terms. In this case,
in a matrix of one value with the same size as the model, the entire extracted section along the seismic profile is weighted
accordingly for all rock units. These spatially distributed weighting sections (matrices) are then transferred to vectors and
applied as global and local constraints to the previously introduced regularization terms. It is important to increase the values
of weights to reduce model updates around the seismic interpretations and to suppress the changes of the interface locations in
low-uncertainty areas.

As the calculation of the sensitivity matrix ($\mathbf{J}^\phi$) is limited to the vicinity of interfaces defined by $\tau$, the search space for
applying the constraints can also be limited to neighbouring cells around interfaces. This indicates that regularizations can also
be adjusted over boundaries between units rather than the whole lithological units. This can be advantageous in cases where
extracting the array locations along the boundaries is straightforward. Detected unit boundaries from other sources of
information (geological and geophysical) can then be directly transferred to weighting matrices to be used in the constrained
level-set inversion. Boundaries' parametrization from interpretations or inversion (such as acoustic impedance inversion) of
seismic data after depth conversion and projection onto the mesh used for the inversion can be directly used as regularizations.
It is then plausible if interpreted boundaries from regional seismic studies or from inversion are digitized as weighting matrices





and used as constraints. Demonstration of the structure of a weighting matrix along boundaries of a given unit of the conceptual model can be observed in Fig. 1b.

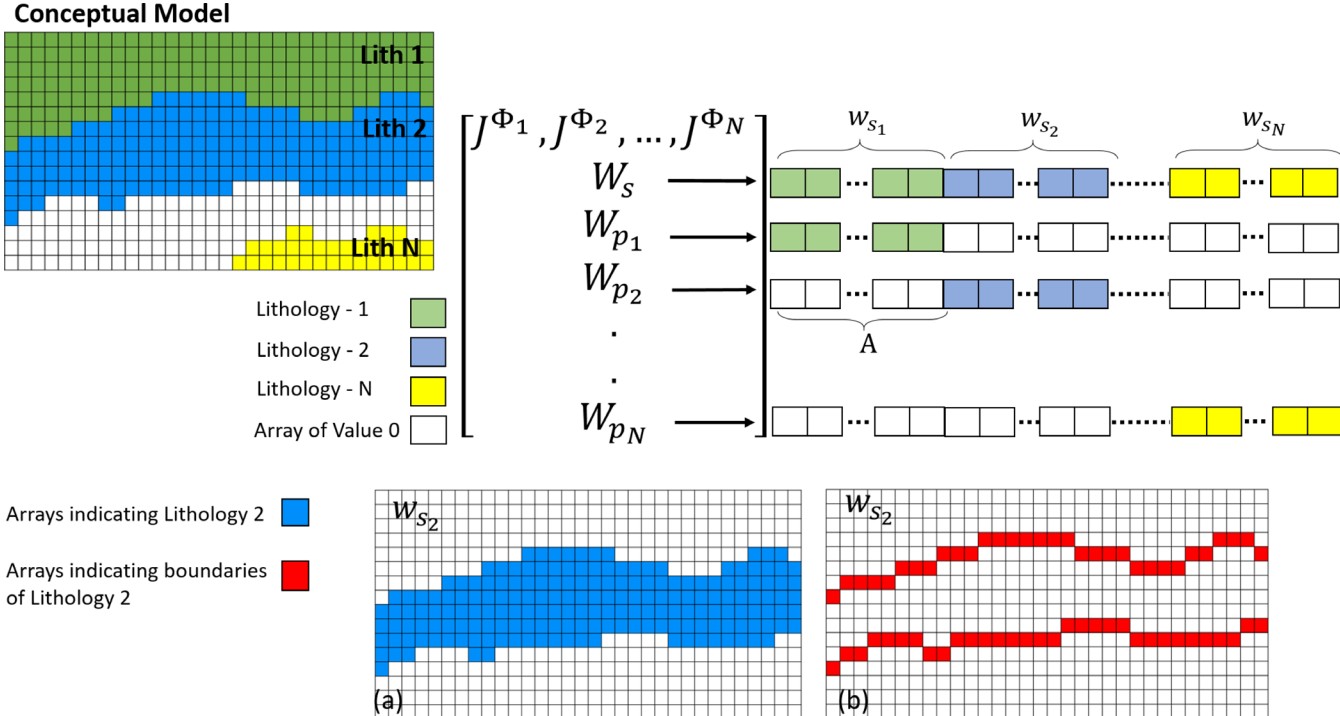

**Figure 1: Illustration of the process of appending constraints to the level-set problem from a sample 2D geological model. Distribution of constraint matrix of from lithology 2 for the entire unit (a) and for the corresponding boundaries (b). $W_{s_2}$ is shown only for the corresponding 2D section.**

As the application of sparse constraints enforces local restrictions to the evolution of signed-distance values, the resulting model from ensembles of signed-distance functions for different lithologies also requires constraints to enforce small-scale topological rules accordingly with geological knowledge and to ensure that certain configurations do not occur during inversion. This is covered in the next subsection.

## 2.4 Enforcing topological rules

Topological constraints play a paramount role in geo-modelling processes (Burns, 1988; Pellerin et al., 2017; Perrin and Rainaud, 2013; Thiele et al., 2016). They refer to properties of a model that are changing during the perturbation of a model. Topological relationships can be defined over discontinuity networks (fractures or faults) (Sanderson and Nixon, 2015), lithological units, or unconformities (Jessell et al., 2010) and are decisive components in 3D modeling process. In recent years, these relationships are computationally being considered either explicitly or implicitly (Pakyuz-Charrier et al., 2018a; Pakyuz-Charrier et al., 2018b) and as constraints for probabilistic 3D models (De La Varga et al., 2019) in the context of 3D geological modelling. Furthermore, in the geophysical inversion context, they have been sometimes addressed by means of uncertainty



quantifications during inversion and joint-inversion problems (Giraud et al., 2019b, 2017; Wellmann et al., 2018) or as post-inversion regularization analysis (Cracknell & Reading, 2015; Giraud et al., 2020; Tarabalka et al., 2009). However, to the best of our knowledge, the application of the topological relationships while deforming discrete units in geophysical inversion problems has not been addressed.

Among different orders of the topology (Thiele et al., 2016) we focus on first-order topology describing adjacency relationships
between rock units. The utilized level-set inversion approach allows us to enforce small-scale topological rules as morphological constraints to the inversion problem. To ensure that inverted models remain geologically realistic, we apply topological rules at each iteration. We take advantage of morphological rules of image processing techniques to prevent the nucleation of a given unit into another and for the model to obey topological rules. This becomes important for retaining the integrity of the predefined unit boundaries during the inversion and geological plausibility (age and deformation history).

Application of the mathematical morphology on geoscientific datasets have been evaluated based on the classification of input data types (Heijmans, 1995; Serra, 1986; Soille and Pesaresi, 2002). However, it has rarely been applied within the geophysical inversion problems. Given that in a level-set inversion problem the model space can be assumed as a multinary image where the ordering of signed-distance values matters more than property values, morphological opening or closing (Vincent, 1993) rules can be applied on the multinary lithology ($\mathbf{b}^{lith}$) at each iteration. For a 3D model, considering a structuring element $\boldsymbol{\gamma}$
of size $(m \times n \times p)_{0<m,n,p<model\ dimension}$ , the closing notation (•) of $\mathbf{b}^{lith}$ can be shown as:

$$\mathbf{b}^{lith} \bullet \boldsymbol{\gamma} = (\mathbf{b}^{lith} \ominus \boldsymbol{\gamma}) \oplus \boldsymbol{\gamma} \qquad\qquad (7)$$

Where $\ominus$ and $\oplus$ demonstrate morphological dilation and erosion respectively (Jankowski, 2006; De Natale and Boato, 2017). This operation ensures that all neighbouring model cells with the size of structuring element that does not fit in the background density, will be closed, thus nucleation of each lithology with size more than $(m \times n \times p)$ is prevented from occurring.

## 3 Synthetic examples

The aim of this section is to study the application of the introduced procedure on two synthetic case studies. We first benchmark the method using a well-known model and then simulate a more realistic application in a hard-rock scenario where 2D constraints are applied in a 3D inversion setting.

### 3.1 SEG/EAGE salt dome model

In this section, we use a simplified version of the SEG/EAGE salt dome model (Aminzadeh, 1996) in the same fashion as Li et al (2016). This example demonstrates the application of the regularized level-set inversion for a model of two distinct rock units. We assume the salt dome in Fig. 2a with the $-470\ kg.m^{-3}$ density contrast in a void background assuming $2670\ kg.m^{-3}$ as the base density. We discretize the model to cells of $200\ m$. The model extends from 0 to 13520 m in both





horizontal directions and from 0 to 4000 $m$ in depth (generating a model volume of size $n_x \times n_y \times n_z = 68 \times 68 \times 21 =$

97104). A cross-section of the model and the corresponding synthetic seismic image is extracted along the oblique line ($t -$

$t'$) in Fig. 2c (from $X_t$=3000, $Y_t$=0 to $X_{t'}$=13000, $Y_{t'}$=11000 m) to image the salt body for the constraining purpose. We also

add 5% noise with normal distribution to the forward calculated gravity datasets to generate the field datasets on the

surface ($N_x \times N_y = 30 \times 30 = 900$). It is assumed that the subsalt boundary and also the dipping part of the salt have been

poorly imaged and are not interpretable from the seismic section (Fig. 2b). This restricts the constraining matrices from the

seismic section only to the upper boundary of salt (Fig. 2d). The constraining matrices are then appended to the sensitivity

matrix as global and local regularizations for both units.

In many of the real case scenarios the starting model although follows the primary assumptions in the region, might be far

from the reality. Thus, we start the inversion using a randomly generated starting model different from the true model to test

and evaluate the similarity of the resulted geometry with the true geometry. In this example, we consider a disc as starting

model (Fig. 2e) that has intersections with two edges of the model boundaries.

The results of the level-set gravity inversion of the salt dome using starting model in Fig. 2e with and without the utilization

of seismic information are illustrated in Fig. 3. The resulted model demonstrated in Fig. 3a is recovered after applying

constraints along 2D seismic section based on the well-imaged upper boundary of the salt. We apply the constraints for this

example along the seismic section using the weighting value (400) along the well-imaged top salt boundary (black cubes in

Fig. 2d). The initialized value for $\tau$ is considered 100.2 $m$. The data misfit error decreases smoothly and converges to 0.8

mGal almost after 4 iterations. Due to the simplicity of the model, the morphological closing is not included during the

inversion of this model.

The number of iterations toward convergence in this example is noticeably low for imaging the salt body for both of constrained

and unconstrained inversion. The model misfit between the resulting inverted model from the constrained inversion and the

true model (99 $kg.m^{-3}$) is slightly smaller than the model misfit resulting from the unconstrained inversion (106.2 $kg.m^{-3}$).

Although the lower boundary of the salt has not been constrained by seismic image, application of the constraints to the upper

part of the salt body has also improved the imaging of subsalt.

The application of the regularized inversion on a more complex model by utilizing regularization terms in a vertical section is

provided in the next section.






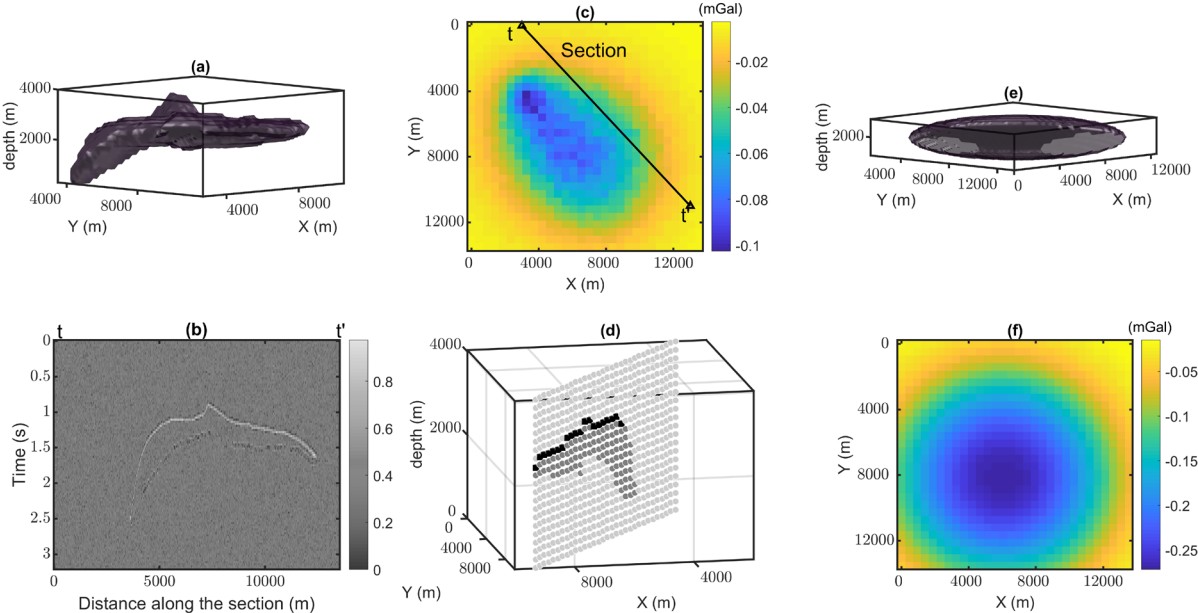

**Figure 2: (a) Simplified SEG/EAGE salt model in 3D. (b) Generated synthetic seismic along section shown in (c). (c) Forward calculated gravity data of the true model on the surface with 5% noise. (d) Extraction of the model along the seismic line to construct the constraining matrix, the black cubes represent the location of indices of the constraining matrix with the maximum weight (400) based on the seismic image. (e) Starting model for the inversion and (f) the calculated forward gravity data of the starting model.**



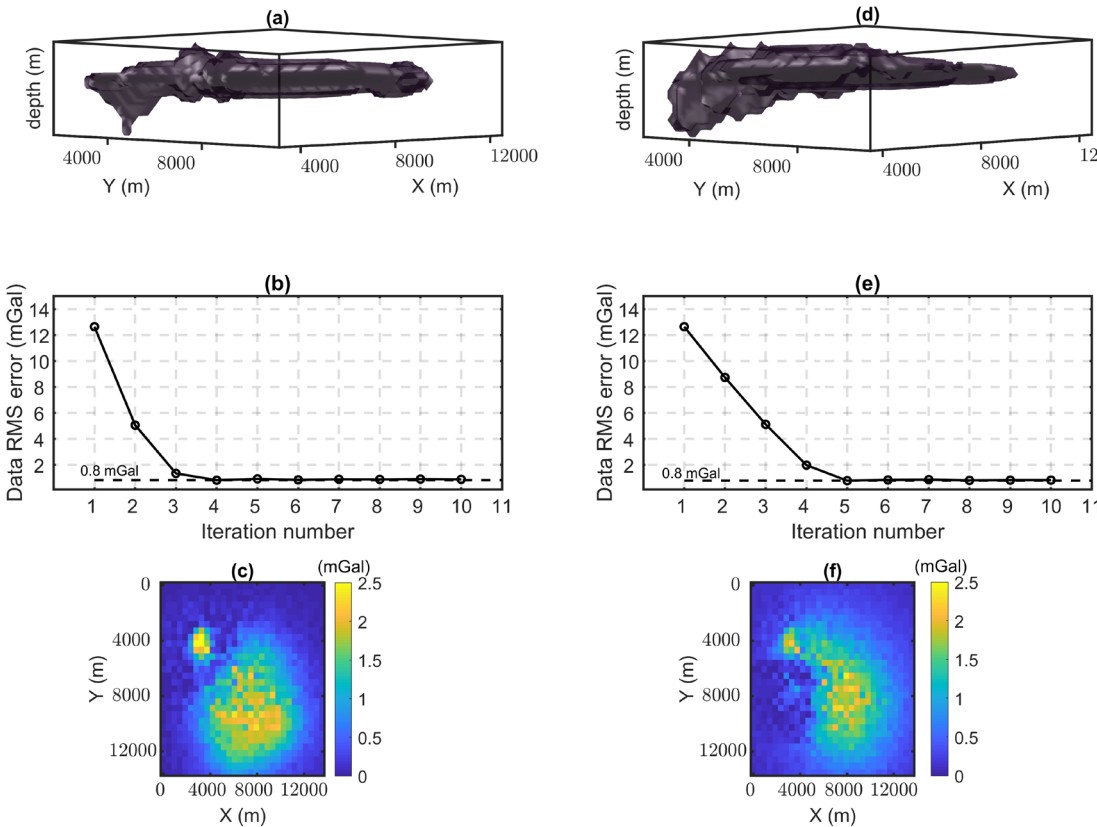

**Figure 3: 3D views of the inverted model from the constrained (a) and unconstrained inversion (d). Corresponding evolution of data RMS errors (b and e), and (c) the corresponding final calculated difference between the field and calculated gravity datasets (c and f).**

### 3.2 Example of hard-rock synthetic case

We have also tested the method on a second generated synthetic model (Fig. 4a) which contains four distinct geological bodies with different density contrasts. The model simulates a hard-rock scenario and contains two exposed bodies (greenstones) with the same density contrast ($330 \ kg.m^{-3}$) surrounded by lower-density geological units (granitic background). This model is considered as the true model and the aim of the 3D generalized level-set inversion is to recover a model that is structurally close to the true model by constraining the inversion using the seismic information available in the area. The model is composed of $n_x \times n_y \times n_z = 40 \times 30 \times 40 = 48000$ cubic model cells with 50 m resolution. A zero-offset straight synthetic seismic section along y = 750 m using a finer grid mesh (10 m each cell dimension) is generated and 2% random noise with normal distribution is added to the amplitudes (Fig. 4c). The true model is then used to simulate gravity anomaly data at surface level ($N_x \times N_y = 36 \times 26 = 936$) with 50 m spacing and assuming padding of 100 m. We add 5% noise with normal distribution to the gravity measurements (Fig. 4b). As the main focus of this study is to test the application of 2D vertical constraints in a





3D level-set inversion, we only consider the seismic section for generating the starting model. For the purpose of testing our algorithm, we simulate a starting model with an inaccurate representation of the geology of the area. In particular, connectivity of the greenstones and differences between the position and dipping of all layers (Fig. 4d). The starting model is used to initialize the signed-distance functions. To add distributed constraints along the seismic section to the inversion, we use the absolute value of reflectivity coefficients multiplied by constant values to directly translate boundaries from the seismic section

to weighting matrices in the level-set formulation in Eq. (6). Translating these boundaries to weighting matrices has been explained in detail in the theory section (Illustrated in Fig. 1b). Reflectivity as a measure of acoustic impedance can be calculated with the knowledge of the wavelet frequency. We use the same reflectivity matrix that was calculated for generating the synthetic seismic. Eventually, we apply weighting factors of 200 on model cells along the seismic section with sharp boundaries as maximum weights along reflectors for the constraining purpose. The morphological closing using a structuring

element of size $(100 \times 100 \times 100 \ m^3)$ is also applied to the resulting model to prevent nucleation of one lithology into the other. The initialized value for $\tau$ is considered $35.05 \ m$. The inverted model stabilizes around an acceptable solution after 6 iterations with a total data misfit of 0.18 mGal (Fig. 5b). Qualitatively, visual inspection reveals that the resulting inverted model is in an acceptable agreement with the true model (Fig. 5a).

To assess the influence of seismic-derived constraints, we repeat the level-set inversion without applying the constraints along

the section. The resulted model of 3D gravity inversion without seismic constraints is illustrated in Fig. 5d, with an overall misfit of 0.10 mGal in data RMS error (Fig. 5e).

A comparison of final models from constrained and unconstrained inversion is presented in Fig. 6. The model differences between inverted models and the true model in Fig. 6c and Fig. 6d show more improvement in the constrained case. Model RMS error shows lower values for the constrained inversion ($72.5 \ kg. m^{-3}$). We also compare the results by measuring the

structural similarity between the inverted models and the true model. A measurement for structural similarity (SSIM) (Wang et al., 2004) that models the perceived changes in structural information of two different models **A** and **B** can be used as an indication of changes in the unit boundaries' location after level-set inversion.

$$SSIM \ (\mathbf{A}, \mathbf{B}) = \frac{(2\mu_{\mathbf{A}}\mu_{\mathbf{B}}+c_1)(2\sigma_{\mathbf{AB}}+c_2)}{(\mu_{\mathbf{A}}^2+\mu_{\mathbf{B}}^2+c_1)(\sigma_{\mathbf{A}}^2+\sigma_{\mathbf{B}}^2+c_2)} \tag{8}$$

$$C_1 = (K_1L)^2, C_2 = (K_2L)^2$$


Where $\mu$ and $\sigma$ are mean and variance of two models respectively, $\sigma_{\mathbf{AB}}$ stands for the covariance of two models, $K_1$ and $K_2$ are two small constant values and $L$ is the dynamic range of the density contrast values. For two identical models, SSIM will become 1. The results indicate that final generated models from seismically constrained gravity inversion has recouped structural features of the true model up to 65%. Although the unconstrained inversion leads to a lower data misfit error, the

structural similarity between the inverted model and true model is less favourable than in the seismically-constrained case (33%). The difference between the structural similarities is an indication of the applicability of the approach to utilize





spatially distributed constraints in the level-set inversion and implies that the method can be applied to real case scenarios where gravity and seismic datasets with different coverages are available.

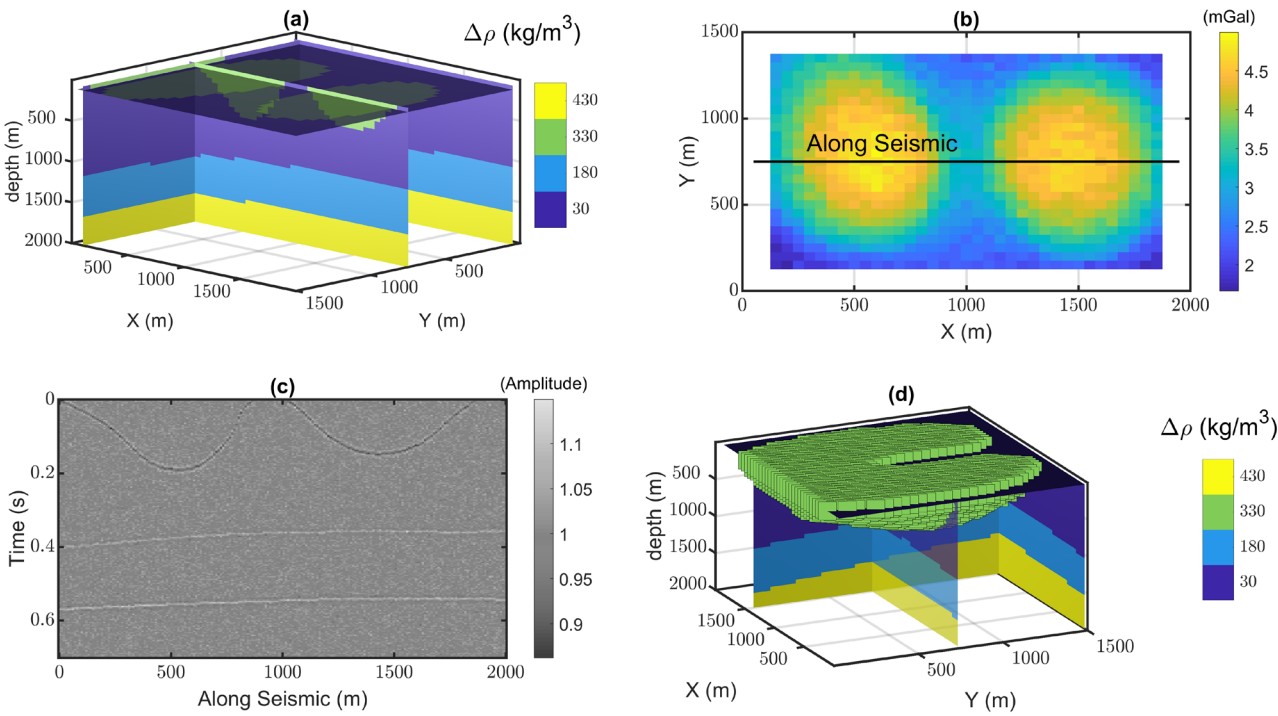

**Figure 4: 3D view of the true model (a) and surface gravity anomaly response (b), zero-offset synthetic seismic section (c), 3D view of the starting model (d).**





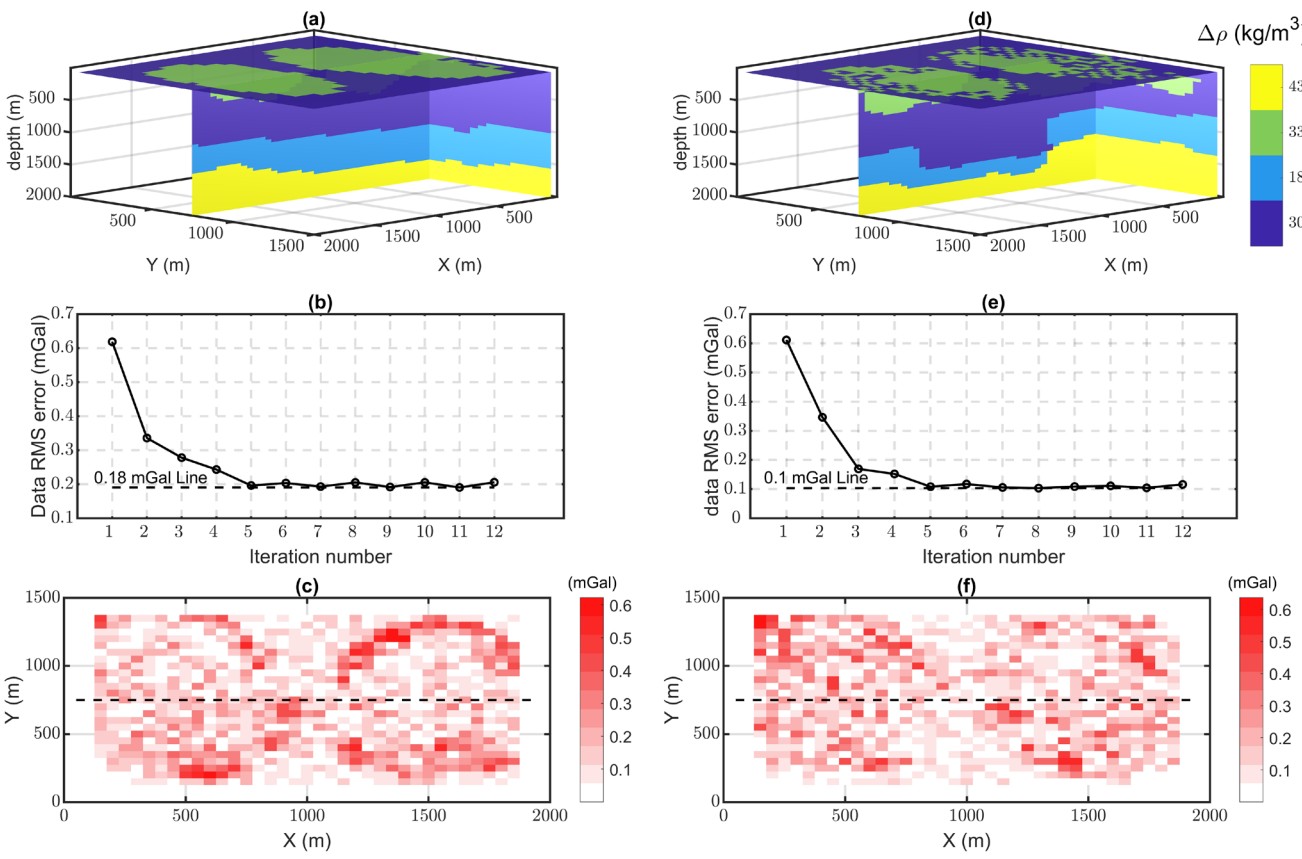

**Figure 5: Final inverted models from the seismically-constrained (a) and unconstrained inversion (d), evolution of the data RMS error during constrained (b) and unconstrained inversion (e) and the corresponding differences between the new data and synthetic observed data respectively (c & f).**





**Figure 6: a) Inverted model from the constrained inversion along seismic section. b) Inverted model from the unconstrained inversion along the same section. c) Difference between model in (a) and the true model. c) Difference between model in (b) and the true model. Below these figures, the corresponding data and model RMS error and SSIM of two models are compared.**




**4 Case study: Yamarna Terrane (Yilgarn Craton, Western Australia)**

**4.1 Geology of the area**

The importance of Yilgarn Craton to the economy of Western Australia is evident as it consists of numerous granite-greenstone terranes hosting world-class deposits of gold and nickel (Whitaker, 2004). Several terranes are defined in this craton based on

geochronological dating of magmatism and further geochemistry analysis (Cassidy et al., 2006; Pawley et al., 2007). The eastern portion of the Yilgarn Craton is the Eastern Goldfields Superterrane which is divided into four terranes: Kalgoorlie, Kurnalpi, Burtville and Yamarna (Pawley et al., 2007). The Yamarna Terrane is located in the northeast (Fig. 7). It is proposed that the Yamarna Terrane evolved separately from the original Burtville terrane based on the different character of volcano-sedimentary events (Pawley et al., 2012). While there are similarities between the Yamarna Terrane and the sequences around

Kalgoorlie (a town host to one of the largest gold deposits in the world), the lack of significant historical mineral discovery is at odds with the apparent prospectivity of the region. Complicated mineral exploration by extensive Phanerozoic cover (Lindsay et al., 2020), significant regolith (Anand and Paine, 2002) and poor outcrop encourage new geophysical investigations in this area to unlock the structure permissive for mineralizing processes (Goleby et al., 2004; Lindsay et al., 2020).





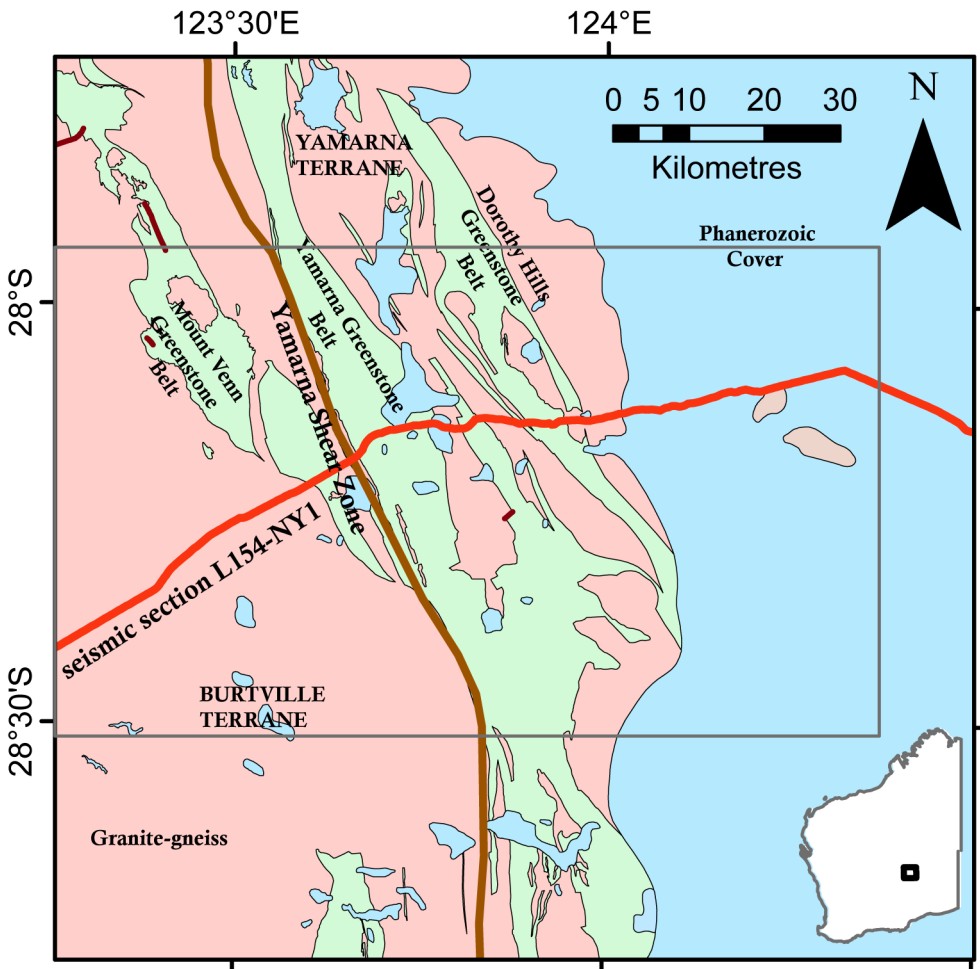

**Figure 7: Geographical coordinates and geological units of the region of interest in the Yamarna Terrane, Eastern Yilgarn Craton, Albany-Fraser Orogen (Modified from Lindsay et al., 2020). The grey rectangular show the boundary of the region of interest in this study.**

Numerous geophysical datasets including regional gravity data and 2D reflection seismic profiles have been collected in this area. However, few studies integrating these data have been done to estimate the 3D structure of the greenstones. The targeted greenstone belts are proximal to a series of major and minor faults and shear zones which make it necessary to utilize geophysical integration for regional studies to have a better understanding of these metal hosts. Gravity datasets, although providing satisfying lateral resolution, are barely capable of estimating the depth of sub-horizontal interfaces. 2D seismic sections, on the other hand, contain information of deep structures that need complementary datasets to be laterally extended.





### 4.2 Geophysical datasets

The region of interest was chosen based on primary interpretation of greenstone locations approximated from the 2D seismic

section and surface geological mapping so that it covers an area where two major shear zones (Yamarna and Dorothy Hills)

and bounded greenstones cross the long seismic section. This region of interest contains more sparse gravity measurements

toward Burtville Terrane to the west and Dorothy Hills shear zone to the east (average 10km spacing) while denser gravity

measurements with 2km spacing are collected around the Yamarna shear zone (Fig. 8). In this study, we use the interpolated

regular grid of gravity anomaly data on the surface (from the Geological Survey of Western Australia) with 400-meter spacing

(easting: 535000 to 625000 m; northing: 6860000 to 6900000 m) including the spherical cap and terrane corrections in the

datasets. The chosen area contains part of the reflection seismic data from common-depth-point (CDP) number 13500 (easting:

539639.00; northing: 6866100.00 m) to 18000 (easting: 622269.00; northing: 6891816.00 m) with an approximate average

CDP spacing of 20 $m$.

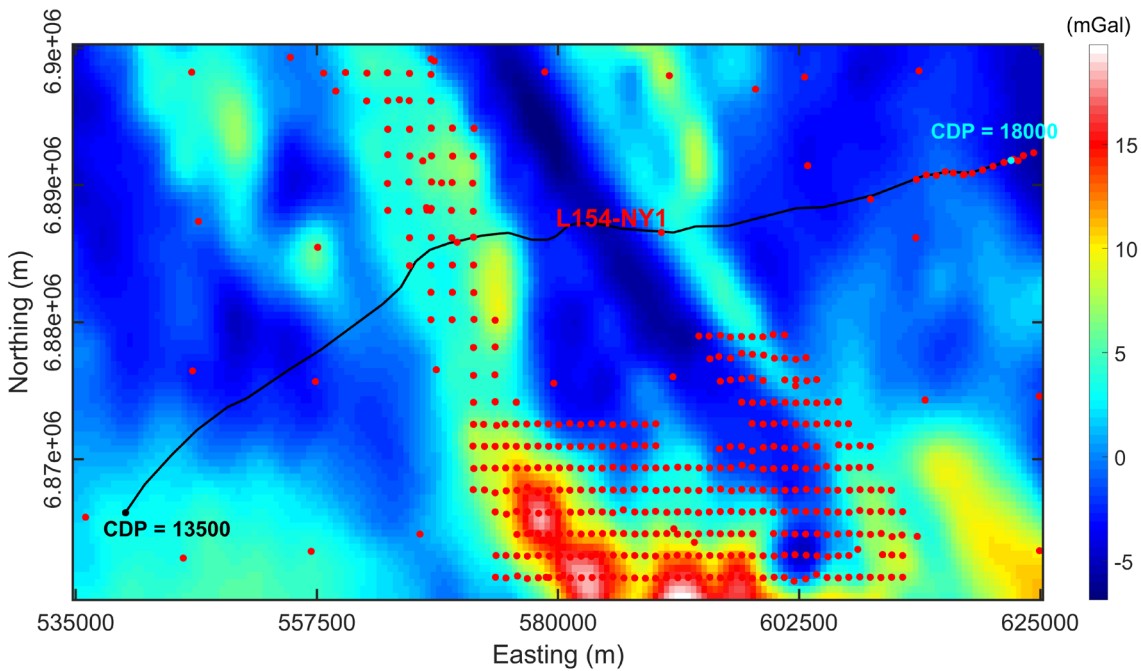


**Figure 8: Interpolated gravity anomaly grid of Yamarna Terrane, from Geological Survey of Western Australia. Seismic traverse is shown as a solid black line within the region of interest (from CDP=13500 to CDP=18000). Red dots indicate the original locations of gravity stations.**

### 4.3 Starting model generation

A general overview of the geological setting of this region based on deep reflection seismic profiling with a focus on Laverton

Tectonic zone modelling is presented in Goleby et al (2004). We use their primary interpretation along the seismic section

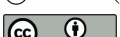



within our region of interest together with modifications based on the integrated multi-scale study presented in Lindsay et al (2020) to generate a starting model for the gravity inversion in the Yamarna terrane.

Referring to primary seismic interpretations from Goleby et al (2004) and Lindsay et al (2020), major shear zones hardly
extend deeper than 8km in depth. Focusing on the upper 8 km, we remove the effects of the regional gravitational trend and recalculate the residual Bouguer anomaly (Gallardo and Thebaud, 2012). Therefore, we generate a model grid from the surface down to 10km depth and discretize to 500 m resolution with cubic cells.

Due to the insufficient number of studies in the region, we mostly rely on the results from a multi-scale study presented in Lindsay et al (2020) to generate the starting model. In this report, diverse geophysical datasets are utilized to conclude a general
integrated interpretation of the region. The presented results from magnetotelluric modelling, reflection seismic reprocessed image and potential field gravity inversion are used for geoscientific investigation. These results from the geophysical modelling are then used to relate petrophysical signatures to geological units using machine learning techniques. We initialize the density contrasts (with the base density of $2670\ kg.m^{-3}$ ) based on a primary gravity anomaly interpretation provided in the report as shown in Fig. 9a. The density contrast values assigned to defined units are inferred from the same report as well
as interpretations from Goleby et al (2004) and summarized in Table 1 (first row). For more details about the integrated interpretation of the region, we refer the readers to Lindsay et al (2020).

The ultimate aim of this case study is to produce a 3D model from the seismically constrained level-set inversion that is consistent with the surface gravity anomaly while including detectable, thus plausible, structures from the 2D seismic section. To achieve this ambitious aim, we first break down the workflow to 3D property inversion followed by the unconstrained and
constrained 3D level-set inversions.

### 4.4 Modifying the starting model

We use the generated knowledge-driven model as the prior model for the density contrast inversion. The Tomofast-x inversion platform (Giraud et al., 2021b; Giraud, et al., 2020; Giraud, et al., 2019a; Martin et al., 2020) is used for the property gravity inversion with smallness constraints from the prior model. The inverted density contrast model and the resulting calculated
datasets together with the seismic overlayed image are shown in Fig. 9b to 9d.

Before level-set inversion, we modified the starting model based on the property inversion results and utilized the resulting model (Fig. 10) for the 3D level-set inversion. The main motivation for modifying the starting model based on the property inversion result is to generate a subsurface model that follows both geophysical datasets and also complements the prior geological knowledge of the region. We tie all this information together and obtain a geologically and geophysically plausible
updated model of the area. In this modified model, we assign the same density contrast values to Mount Venn and Yamarna greenstones and consider them as a single density contrast unit as suggested from the property inversion results. In addition, we propose an alternative scenario where we added another density contrast unit to the west as derived from the property





inversion result. Primary 3D level-set inversion of the region of interest using the initially defined starting model strongly

suggests the generation of a new geological unit to the west with different dipping from the general east-dipping structure of

the area. This appended unit to the west can also be inferred from the seismic image (Fig. 9d) where there are some flat-dipping

heterogeneities toward the west edge. We assign a data-driven density contrast heterogeneity to the unit toward the west of the

model based on the property inversion results. The ultimate density contrast values used for generating the modified starting

model are summarized in Table 1. This modified model is then used as the starting model for the 3D level-set inversion of the

region forming five distinct level-set functions. The traversed seismic section in the region of interest covers all five defined

rock units in the region (Fig. 10b). Furthermore, the modified starting model along this line is utilized for constraining purposes.

Out of convenience, we use the section view of the model along the seismic profile for most of the visualization purposes.

The discussion over testing different ranges of density contrast values for different rock units is beyond the scope of this study.

However, several sets of density contrasts based on generating a similar range of forward gravity datasets with the field datasets

have been tested. Due to recovering almost the same structures by using different density contrast ranges, we present the results

of one set of the resulting geometries (using densities in the second row of Table 1) in the next sections.

**Table 1. Density contrasts assigned to different rock units (Units are in $kg.m^{-3}$).**

| Rock Unit | Mount Venn | Yamarna | Dorothy Hills | Lake Yeo | Background | Toward West |
|---|---|---|---|---|---|---|
| **Starting** | 490 | 530 | 470 | 50 | 20 | - |
| **Modified** | 120 | 120 | 80 | 100 | -70 | 20 |



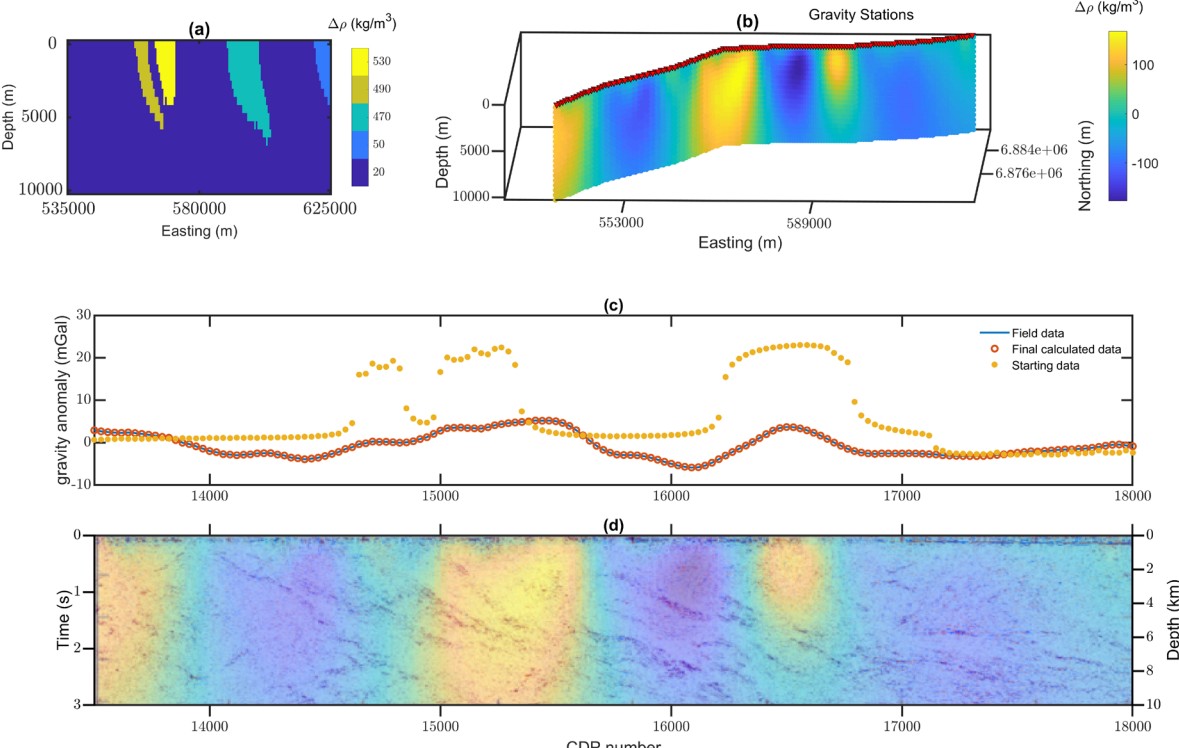

**Figure 9: (a) Prior model for the gravity property inversion (b) gravity density contrast inversion result along seismic profile (c) comparison of the field data and calculated gravity anomaly from the final inverted model (d) overlayed gravity inversion result on seismic profile.**

### 4.5 Level-set inversion in 3D

The aim of this section is to illustrate the application of the gravity level-set inversion algorithm to constrain the 3D density model of the area with the seismic section. The starting model for the 3D level-set has five distinct units with densities as shown in Fig. 10. The model size for the 3D inversion is $n_x \times n_y \times n_z = 180 \times 80 \times 20 = 288,000$, and the data size is $N_X \times N_Y = 50 \times 20 = 1,000$. As stated earlier in this section, very little is known about the area and the generated starting models are highly uncertain. Therefore, the results presented in this section are also undetermined and need to be interpreted with caution. The calculated gravity anomaly responses from the starting model and a comparison with the field datasets are shown in Fig. 11. A very high misfit between the observed and field datasets indicates that existing interpretations are far from reality.





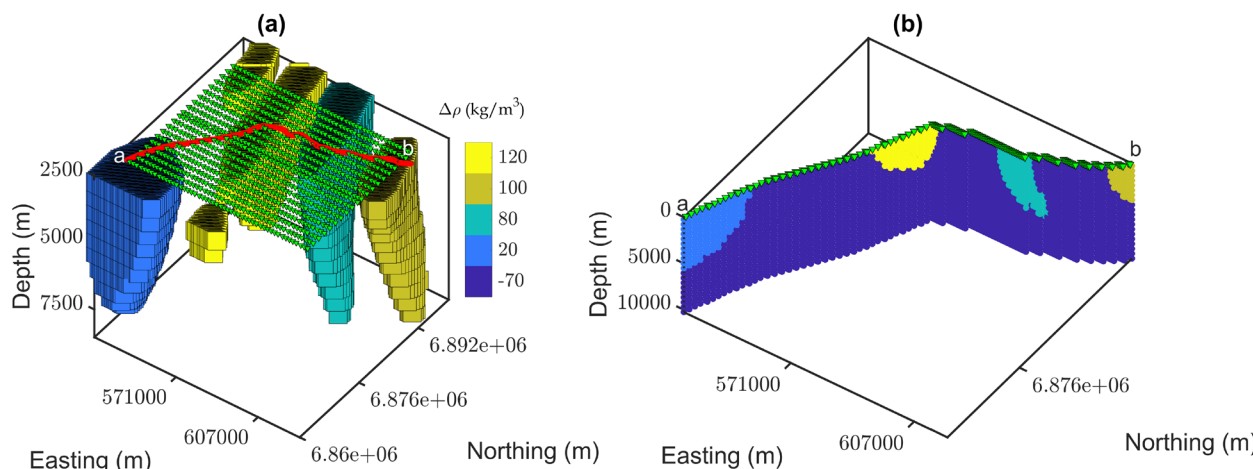

Figure 10: Density contrasts of the adjusted starting model and the interpolated gravity grid configuration (a) model cell centres of the extracted 2D model along the seismic profile (b).

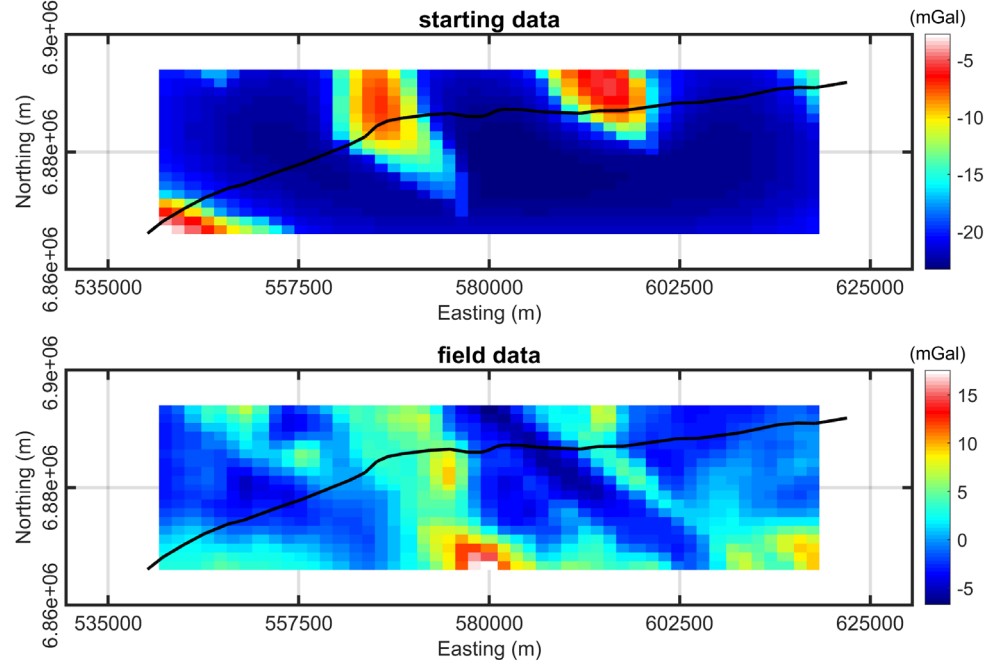

Figure 11: The starting and field gravity datasets of the Yamarna Terrane. Solid black lines represent the seismic traverse.





### 4.5.1 Unconstrained 3D level-set inversion

We first implement the level-set inversion without applying constraints along the seismic profile to compare the resulting model from level-set inversions without and with seismic information. In this scenario, uniform local and global regularizations vectors are appended to the sensitivity matrix without including the weighting from seismic information. We set all regularization parameters to 1 except for the layer at the surface level where we assign a small number (500) to the weighting terms to include information from geological mapping. Morphological closing is also applied as previously introduced using a structuring element of size $(400 \times 400 \times 400\ m^3)$. The depth weighting term for compensating the decay of the gravitational force with depth is also added to the inversion problem as introduced and utilized in Boulanger & Chouteau (2001). The data RMS error of the unconstrained level-set inversion decreases smoothly converging at 1 mGal after 13 iterations (Fig. 13a). The small number of iterations for reaching the optimum value of the data RMS error is due to choosing a relatively broad interface between rock units ($\tau = 2100\ m$). We previously discussed the importance of this parameter to control the search space of the problem in the theory section. Choosing such a large value for $\tau$ allows converging rapidly given the size of the model.

The geometry of the density units of the resulted model however is not in agreement with the existing geological map and seismic interpretations. The final modelled geometry of the greenstones from the unconstrained inversion is shown in Fig. 12a. These geometries are terminated near the surface and are not extended to depth. This is mainly because we do not apply the constraints at depth. Therefore, the application of the regularizations from the seismic profile to improve the integrity of the units as they evolve laterally (as observed in the interpolated gravity grid) and vertically (as interpreted from the seismic profile) is necessary.

### 4.5.2 Constrained 3D level-set inversion

Next, we apply the spatial constraints along the seismic section to constrain the gravity inversion with the primary interpretation along seismic. We aim at reconciling gravity inversion with seismic interpretation. We use the same morphological constraints and depth weighting as we set for the unconstrained inversion in order to provide a fair comparison with the unconstrained inversion result. We also use the same value for the $\tau$ parameter. The updated regularization terms with weighted constraints from the seismic information along the vertically extended profile are then applied to constrain the inversion. All arrays of $\boldsymbol{W}_s$ vector are set as one except along the seismic section and for the layer at the surface level which we set as 2000 and 500 respectively, to suppress the changes of the signed-distance function and to constrain the inversion problem with spatially distributed regularizations. The weighting constants are tuned manually during the level-set inversion. As a rule of thumb, these constants are chosen between double to fourth of the cell size. Repeatedly, starting from a significant data RMS (more than 20 mGal), it takes 12 iterations (Fig. 13b) for the data RMS error to reaches the minimum value (1.4 mGal). The final modelled geometry of the greenstones from the constrained inversion is also shown in Fig. 12b.





**Figure 12: Different views of the final inverted models from the unconstrained and constrained 3d gravity level-set inversion of Yamarna Terrane respectively. Black dot points at the surface level represent the interpolated gravity data.**





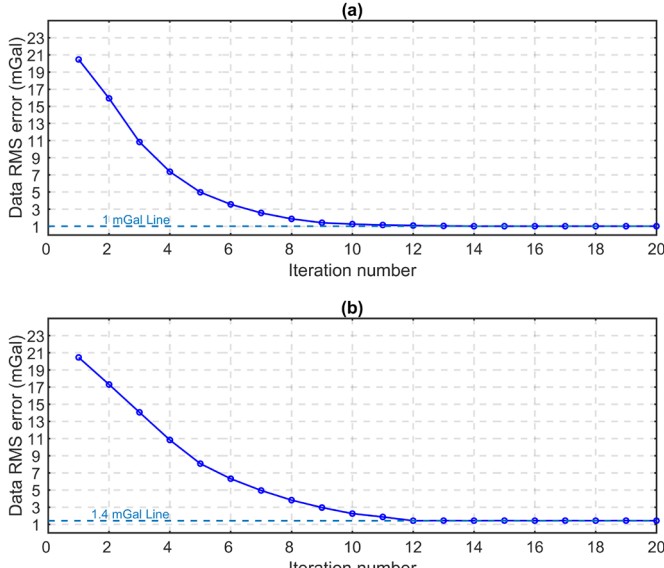

**Figure 13: Evolution of data RMS error for the unconstrained (a) and constrained (b) 3D level-set inversion of Yamarna Terrane.**

Comparing the final model from the inversion (Fig. 12b) and the prior model (Fig. 10), we can observe that the geometry of greenstones along the seismic profile is well-constrained and the surface features also align with the observed gravity datasets

445    (Fig. 14). Most parts of the inverted model are in line with the existing assumptions in the area and represent an updated geometry for greenstone units adjacent to the shear zones. There are differences between these two models along the seismic section while the general trend of features away from the seismic profile for both constrained and unconstrained scenarios show very similar patterns. This is an indication of the capability of the approach for applying the constraints along a vertical section and constraining the gravity inversion with sparsely distributed information within the model.




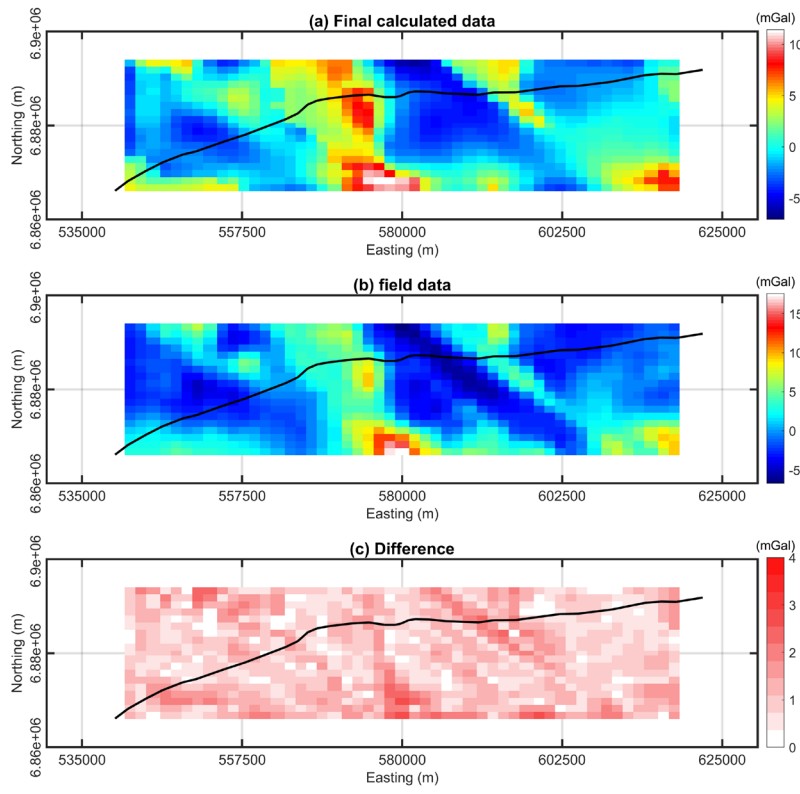

450

**Figure 14: (a) Calculated data from the final constrained inverted model (b) field data and (c) the absolute difference. Black solid lines represent the trace of seismic profile.**

Figure 15a displays the results of the constrained inversion and recovered geometries along the seismic profile with the interpretation of Goleby et al (2004) overlain. As demonstrated in Fig. 15 the recovered geometry of the greenstones can be correlated with some detectible features in the seismic image which were not primarily specified. We also provide a comparison of the recovered model with the integrated interpretations provided in Lindsay et al (2020) in Fig. 15b.

The source of the difference between these interpretations is mainly because in Goleby et al (2004) the focus was more on the whole of crust, whereas Lindsay et al (2020) were looking nearer to the surface and also utilized the re-processed seismic profile in their study.



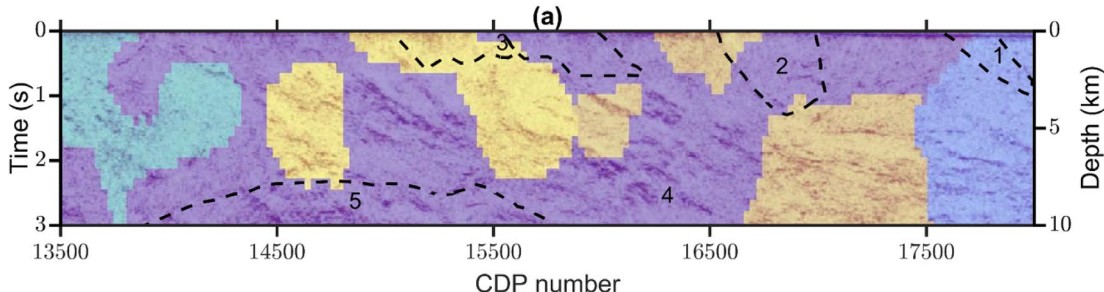

1, 2, 3: Greenstones (mafic/Ultramafic);   4: Granite/Gneiss Background;   5: Basement

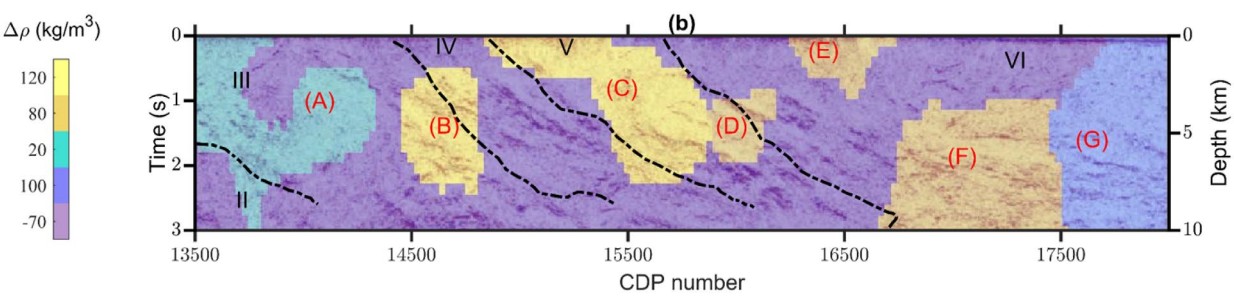

---- : seperators for zones I to VI (from Lindsay et al (2020))

**Figure 15: (a) Overlayed final inverted model from the constrained 3d level-set inversion with the seismic image and existing primary interpretations in Goleby et al (2004) (Zones 1 to 5). (b) Overlayed model with integrated interpretations based on seismic characters in Lindsay et al (2020) (Zones I to IV). Notations in red (from A to B) indicate distinct units in the resulting model from the constrained inversion.**

## 5 Discussion

The main focus of this study has been to enable appending spatially distributed constraints in 2D from seismic section to 3D gravity inversion. We have shown this capability across two different synthetic models and a case study to demonstrate different workflows for constraining purposes depending on the available datasets and the area of study.

As detailed in the introduction, seismically constrained gravity inversion has been addressed for a long time. However, constraining gravity inversion quantitatively with sparse seismic information in a cooperative workflow using the level-set technique is novel. While we have used information from reflection seismic as spatially distributed constraints, we see no limitation of using the same approach with other geophysical techniques that provide spatially distributed constraints within the model. For instance, any boundary recognition of rock units from passive seismic, magnetotelluric or electromagnetic techniques can also be translated to weighted constraints to the level-set inversion in the same fashion as we used for reflection seismic.



The results of the first synthetic study using the salt dome model were presented to demonstrate the application of the approach on a simple model where information from the seismic data is incomplete. The results from the salt dome scenario confirm the applicability of the technique on a simple model where the starting model is far from reality and the constraints are not perfect.

A plausible image of the lower body of the salt dome is revealed after level-set inversion for both the constrained and unconstrained examples, thus makes a plausible case for using this level-set technique to image subsalt bodies. The small number of iterations toward convergence using this method is considerable. This shows that the approach is computationally efficient to be integrated with other approaches (seismic imaging techniques). It can compensate for the poor imaging of salt structures which has been an issue in the petroleum industry for a long time and is very important in terms of hosting volumes

accumulation of oil and gas resources. The results on the second and more complex 'hard-rock' synthetic model mostly focus on demonstrating the applicability of the technique where more numerous rock units and density contrasts are required. In contrast to the salt dome example, the starting model is closer to the 'true' model and allowed closer observation of vertical constraint effects.

The level-set method is applied to a part of the eastern Yilgarn craton, a real-world case study where very little is known about

the larger-scale 3D geology of the area. Testing revealed that the constraints generated from the 2D seismic profile and applied to the 3D level-set inversion were effective and provided an informative update to the model of the subsurface. The results do not completely disagree with existing crustal interpretations presented in the literature (Goleby et al., 2004; Lindsay et al., 2020). We note that, based on our results, some new and modified features could be appended to complement existing interpretations.

From CDP number 13500 to 15000 to the left of the section in Fig. 15, the granite-gneiss outcrop is the main detectable feature on the surface. However, we have assigned a small density contrast value (based on a small gravity anomaly) which also agrees with the moderate east-dipping structures in the seismic image (Unit A in Fig. 15b). The main detectable structure from the seismic image is almost compatible with the geometry of the rock units assigned to Yamarna Greenstone Belt (unit C) which in our resulting model is extended by unit D (a unit with the same density contrast as assigned to Dorothy Hills). This almost

matches with the steep east-dipping structure defined in zone V from Lindsay et al (2020). Toward the right side of the section after Yamarna Greenstone Belt (from CDP: 16700 to 18000 and from 5 to 10 km), there have been minor interpretations that could explain the flat-lying structures of the seismic image. In the resulting model, these structures are assigned to unit F and unit G with the same density contrast assigned to Dorothy Hills and Lake Yeo respectively.

The main detectable feature from the constrained level-set inversion as presented in Fig. 12b is the extension of the density

contrast that we assign to the Yamarna Greenstone Belt unit that displays a keel-like shape with a dip to the southeast. The western edge of the feature also results in strong reflectors in the seismic image (Units C and D in Fig. 15) and has been interpreted to be the Yamarna Shear Zone (Goleby et al., 2004; Lindsay et al., 2020). Our result indicates that the predicted and modelled Yamarna shear zone aligns with density contrast patterns that follow anomalously high gravity response laterally





away from the seismic survey trace. The results suggest that the greenstone belt extends, along with the shear zone, up to 8
km to the north and south and likely beyond the geographic boundaries of the model.

The other interesting feature from the inverted model also in strong agreement with surface geological maps and geophysical
information is the depth extension of the Dorothy Hills Greenstone Belt (Units E). Due to the narrow width of this greenstone
belt on the surface, the extension of this unit toward deep parts of the model is unlikely. Referring to studies that address
greenstone structures in Canada (Thurston, 2015) and Australia (Blewett et al., 2010; Gallardo & Thebaud, 2012) the
assumption is that the width of the greenstone belts are indicative of their depth which explains the shallow depth of unit E.

A disconnected extension of this unit toward the eastern part of the model together with the density contrast that we assigned
to the Lake Yeo unit have formed volumes with high density contrast (Units F and G). This also supports the existing
interpretation provided in Lindsay et al (2020) where they assign a higher density domain to the eastern side of the Dorothy
Hills shear zone.

While our level-set inversion results strongly suggest the creation of a density contrast unit with a different dipping structure
(Unit A) to the west of the model there is a lack of evidence in the literature about the extent of such a density contrast unit.
This is mainly because toward the west edge of the model there is a granite-gneiss outcrop and surface geological evidence
(Fig. 7, toward Burtville Terrane) fails to confirm the existence of such density contrast in depth. Knowledge about physical
properties in a hard-rock environment is not enough to constrain the geological concepts and the effects of other processes
should also be considered (Dentith et al., 2020). This recovered unit, be it a data-driven unit from observed gravity datasets
lies in an area with lower density contrast and higher magnetic susceptibility domain based on presented results of gravity and
magnetic inversion in Lindsay et al., (2020). These changes in density and magnetic susceptibility might not necessarily
support the introduction of new geological unit and could be an indication of some secondary geological processes (Dentith et
al., 2020; Saltus and Blakely, 2011; Whitaker, 2004) that results in local bulk heterogeneities regarding the complex mineral
compositions around this region.

Opportunities for improving the presented level-set procedure are centred on interpretation and model uncertainty. Firstly,
there is considerable uncertainty regarding the appropriate velocity model used during the seismic imaging process utilized
within the level-set algorithm. Some existing potential field geophysical inversion methods (Giraud et al., 2019, 2018) use
uncertainty as a constraint, and integration with reflection seismic data would help resolve more complicated scenarios
involving sparse data, limited petrophysical contrast and inadequate model assumptions. In structurally complex areas such as
in hard-rock scenarios, there are high uncertainties regarding the local and regional interpretations as well as in seismic imaging
techniques. This indicates the necessity of quantified inclusion of uncertainty regarding the interpretation and imaging process
of seismic data while constraining the level-set inversion of potential field datasets. It suggests the development of a new
technique that enables interactive constraining of the potential field and reflection seismic datasets within a level-set algorithm.
One advantage of enabling this type of integration in the level-set framework is the compatibility of the results with implicit

geological modelling. Seismic interpretations and gravity datasets are both very common datasets for generating 3D models of an area while there is no technique that allows including seismic uncertainty within 3D integrated implicit modelling algorithms.

The presented results on different datasets revealed that constraining level-set inversion with sparse constraints from low-uncertainty datasets can effectively improve the (geological) plausibility of the results. Adding constraints to the level-set inversion seemed to be effective in all scenarios and is even more obvious with increasing complexities in the models. Sparse constraints, which we see no limitation for their source, force the inverted models to retain the available prior knowledge of the area to a high degree. This introduces a resulting model incorporated from several sources of geological and geophysical information and is a step toward automating the process of 3D geo-modelling.

## 550 6 Conclusion

We have presented the utilization and extension of a generalized level-set approach for seismically-constrained gravity inversion across different scenarios. The flexibility of the level-set approach we followed allowed us to append 2D constraints from spatially distributed seismic information to 3D level-set gravity inversions. We tested the method on two different synthetic models with different scenarios and levels of complexity. The proposed technique was then applied to the 555 geologically complex Yamarna terrane and an updated model of the area with supporting discussions was provided. The proposed approach has proven to be reliable to quantitatively include sparse constraints with low uncertainty to the level-set inversion. In addition, there is considerable flexibility regarding the constraints which makes the approach widely applicable for integrating with other geophysical datasets and geological models. The availability of other sources of information in the studied area about the depth of interfaces in the future can also be utilized in this framework for possible modifications of the 560 recovered model. Also, any improvement in the interpretation of the seismic profile based on future evidence can be directly injected into the inversion to improve the final recovered model.

## 7 Data availability

The utilized datasets and presented results in this manuscript can be found in Rashidifard et al (2021), (https://zenodo.org/record/4747913#.YJnozbX7SHs).


## 8 Competing interests

The authors declare that they have no conflict of interest.



## 9 Author contribution

MR designed the methodology and performed all modelling and inversion presented in the manuscript with the continuous assistant of JG. MR is the main writer of the manuscript which was redacted with the support of all co-authors. ML assisted in the generation of the 3d geological models of Yamarna terrane and interpretation of the results. MJ provided guidance and supervision while the project was being carried out. VO assisted with the mathematical notations in the methodology section. The manuscript is part of the MR PhD project supervised by the rest of the authors.

## 10 Acknowledgment

The work has been supported by the Mineral Exploration Cooperative Research Centre whose activities are funded by the Australian Government's Cooperative Research Centre Program. This is MinEx CRC Document 2021/xx.

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
