# Peer review of "Constraining 3D geometric gravity inversion with 2D reflection seismic profile using a generalized level-set approach: application to Eastern Yilgarn craton"

_Solid Earth, 2021_

## Referee Comment (RC1)

My comments are organized following the paragraph of the paper

I discover the level-set method with this paper and I had a hard time understanding it from the manuscript. I had then to go through the reading of several papers before entering the manuscript.
On one hand, it's ok, these reading are necessary for learning this method from scratch. On the other hand, the **method** paragraph of the manuscript turned to be of no help to understand it.

2.1 **Generalized level-set method**

I thus had two readings of the summary of the method. I) Reading as a novice. (that I was), in that case this part is just totally incomprehensible; ii) reading as an expert (that I'm almost now…), the part is still confusing and do not contain the important information. In both cases I felt quite frustrated.

The authors have the choice, i) either they consider that Giraud et al. 2021 paper (referred below as G21)  is a mandatory reading, and then remove the method summary, or ii) they give the reader enough material to keep reading the paper before reading G21, if necessary. I think the second solution is the correct one, and without increasing the size, then can give a clear, synthetic description of the problem settings.

I suggest relying on figure 1 that is quite clear and replacing the present method part by:
-   Starting with a geometrical description: medium is discretized by cell/nodes (unclear); model is defined by different geological/geophysical units with boundaries (defined on the same mesh); properties are kept constant in geological units; Hence, define N, M, the scope of phi_k

-   Boundaries; recall in few words and/or reference level-set  and signed distance. A simple drawing showing a 1D phi_k across a boundary with a true versus "smeared" Heaviside would help. Explain what the authors means by a "multinary structure" or leave that to a reading of G21. I thing that eq. 7 and/or 8 of G21 is worth being recalled here.

-   Setting of the inverse problem.
    Eq 1 alone can be misleading. It is worth recalling that it comes from a linearization of the problem. I didn't find the information about the iterative scheme that is used to solve the non-linear problem, I guess it's a steepest descent.

Below are some remarks about the text:

1)  You use throughout the text the notion of "rock unit". It seems to me that "Geological or geophysical unit" would more appropriate since you can deal also with sand, clay, salt, etc.

    line 83:  you introduce signed-distance values to interface calculated by FMM. Without further explanation, this sentence is totally incomprehensible. Outline is inappropriate, use boundary or interface instead.

2)  Ligne 85: the sentence where you transform a "signed distance" to a multinary structure (???) using a smeared-out Heaviside is obscure.

Line 91-102: This paragraph is very confusing and for me incorrect.
The sentence "Initializing the model space…" is confusing. **m()** is the model function that links the modeled data to the parameters, through the signed distance $\Phi_k$. It is not a space, neither in a mathematical sense nor in geometrical sense. And you do not "initialize" a model, unless you talk about the initial (trial) model, you "define" it. You'd rather stick to G21 formulation in this part.

Eq 1 is totally confusing since it mixes a general and an iterative formulation. What is $d^{calc}$? It is never defined.
I suggest to rewrite this paragraph according to a more standard way of presenting inverse problems:

a) You are interested in solving a discrete inverse problem whose direct formulation is:
**d**=**g**(**m**); **d**= data; **m** parameter to be inverted; **g**() the direct function, non linear in our case.
b) You decide to solve this non linear problem using a gradient type method base on a 1st order Taylor expansion

$$\mathbf{g}(\mathbf{m}) \approx \mathbf{g}(\mathbf{m}_0) + \frac{\partial \mathbf{g}}{\partial \mathbf{m}}\bigg|_{\mathbf{m}_0} \left(m - m_0\right)$$

c) Considering the parameters of your direct problem: $\mathbf{m} = \mathbf{m}(\Phi, \rho)$ in which density is kept constant, this turns into:

$$\mathbf{g}(\mathbf{m}) \approx \mathbf{g}(\Phi_0) + \frac{\partial \mathbf{g}}{\partial \mathbf{m}} \frac{\partial \mathbf{m}}{\partial \Phi}\bigg|_{\Phi_0} \left(\Phi - \Phi_0\right) \Leftrightarrow \mathbf{g}\left(\mathbf{m}(\Phi)\right) = \mathbf{g}(\Phi_0) + \mathbf{J}^{\Phi} \delta\Phi$$

d) And you decide to iteratively minimize in a least square sense:

$$\Psi_{i+1}^{r} = \left\| \mathbf{d}^{obs} - \mathbf{g}(\Phi_i) - \mathbf{J}^{\Phi_i} \delta\Phi \right\|_2 \text{ where now } \mathbf{d}^{calc} = \mathbf{g}(\Phi i) \text{ is defined as the result of}$$

the direct problem at iteration i.

Please note that compared to your eq1, I have a sign difference. You never use the residuals r that is defined in line94, is it necessary?

**2.2 Regularization level-set inversion**

Sentences in lines 118-121 are confusing and the statement is incorrect, this regularization does not "*encourage the* $\delta\Phi$ *update to reach specific values stored in q*", but it does "*encourage the product* **W** $\delta\Phi$ *update to reach specific values stored in q*" which is quite different (imagine that W is a Laplacian, or a smoothing operator). Since at this point neither **W** nor **q** are defined, it is difficult to understand what the authors mean.
I suggest that the authors replace the text that is too general by more precise details that are given later in the text.
What is the exact size of **q** vector?

Do you try to impose something like $\left\| \begin{array}{c} \mathbf{W}_S \delta\Phi \\ \mathbf{W}_P \delta\Phi - \mathbf{v} \end{array} \right\|$ minimum?

Besides, why do you mix these two constraints simultaneously?

What is the difference between imposing eq 4 rather than $\left\|\mathbf{W}_{S}\delta\Phi\right\| + \left\|\mathbf{W}_{P}\delta\Phi - \mathbf{v}\right\|$ minimum? or $\left(\delta\Phi - \delta\Phi_{prior}\right)^{T} C_{\Phi}^{-1}\left(\delta\Phi - \delta\Phi_{prior}\right)$?

Should we interpret $W_p$ as a geometrical mask (rather than a weighting) that allows fixing some specific values of boundaries in the different geological units?

**2.3**
The sentence on line 149 is incomprehensible, and the full paragraph from 148-153 confusing.

**2.4** Your explanations are ok, however it is difficult to grasp the influence of this topological rule enforcement on your results. Could you comment on the effects of this processing on the synthetic case for instance?

**3.**

Figure 5 caption: what do you mean by difference between "new data and synthetic", what are the new data?

A general question: in 4.3 and 4.4 you choose to build a starting model from the inversion of density only, then invert for the interfaces only in a second step. Why don't you try to invert simultaneously for interfaces and density values in the different units?

Line 384: "due to … sections". Use a direct formulation instead: "We present … because…"

**4.5.2**
I do not understand which geometrical constraints you apply from the seismic profile. On figure 2d for the synthetic case, we clearly see that your constraint follows the geometry of the reflector. What about results obtained on figure 12? There are no clear reflectors such as those of synthetic examples, but rather several general eastward dipping trends. Which constraints do you apply? Could you provide a plot of these constraints along the 2D section?
You mention on line 459 that Goleby et al. (2004) and Lindsay et al.(2020) use different seismic profiles. On line 341 you mention that you use Goleby interpretation. Is your seismic profile coming from the 2004 study or the 2020 study?

---

## Author Comment (AC1)

**Reply to Anonymous Referee # 1 comments**

Dear reviewer,

We appreciate your constructive feedback and detailed comments. These have helped to significantly improve the manuscript. We address your comments point-by-point below. Our revised manuscript will cover these aspects.

Sincerely, Mahtab Rashidifard, Jeremie Giraud, Mark Lindsay, Mark Jessell, Vitaliy Ogarko

My comments are organized following the paragraph of the paper:

**RC1\_1:** I discover the level-set method with this paper and I had a hard time understanding it from the manuscript. I had then to go through the reading of several papers before entering the manuscript.

On one hand, it's ok, these reading are necessary for learning this method from scratch. On the other hand, the **method** paragraph of the manuscript turned to be of no help to understand it.

**Reply:** Thank you very much for listing a very good improvement point here. We were very much pleased to improve the methodology section based on your comments. We agree that the methodology section is not describing the procedure in detail. In the method section, we rely a lot on the given references and on G21 in particular. We have assumed G21 as a compulsory reading and referenced this work several times within the manuscript. However, we have made changes to the theory section in the revised manuscript to make it clearer based on the following detailed comments in this document. These changes can be mainly seen in the method section from lines 79 to 113. We have also added Appendix A to the manuscript which includes required equations for a better understanding of the methodology section.

**2.1 Generalized level-set method**

**RC1\_2:** I thus had two readings of the **summary of the method**. I) Reading as a novice. (that I was), in that case this part is just totally incomprehensible; ii) reading as an expert (that I'm almost now...), the part is still confusing and do not contain the important information. In both cases I felt quite frustrated.

**Reply:** Thank you for pointing out that the methodology section is not sufficiently explained. Your point is very correct. The level-set method introduced in this paper as is pointed out by the second referee is an adaptation of the generalized level-set approach (by Giraud et al., 2021) for utilising gravity and seismic datasets with different coverage. Thus, in the original submitted manuscript, we didn't go through details in the methodology part. In the revised manuscript, we have taken your suggestions and modified the theory section to make this section more comprehensible.

**RC1\_3:** The authors have the choice, i) either they consider that Giraud et al. 2021 paper (referred below as G21) is a mandatory reading, and then remove the **method summary**, or ii) they give the reader enough material to keep reading the paper before reading G21, if necessary. I think the second solution is the correct one, and without increasing the size, then can give a clear, synthetic description of the problem settings.

I suggest relying on figure 1 that is quite clear and replacing the present method part by: - Starting with a geometrical description: medium is discretized by cell/nodes (unclear); model is defined by different geological/geophysical units with boundaries (defined on the same mesh); properties are kept constant in geological units; Hence, define N, M, the scope of phi\_k

- Boundaries; recall in few words and/or reference level-set and signed distance. A simple drawing showing a 1D phi\_k across a boundary with a true versus "smeared" Heaviside would help.

Explain what the authors means by a "multinary structure" or leave that to a reading of G21. I thing that eq. 7 and/or 8 of G21 is worth being recalled here.

**Reply:** We'd like to thank you for the clear suggested description for the methodology summary section. Selecting option number (ii), with the suggested order, after re-reading the method summary section we find it more graspable. Section 2.1 of the manuscript has been modified based on the three suggested steps above. We have added a new Figure (Fig. A-1) showing the  $\phi_k$  across the boundary and also added the smeared Heaviside function equation in Appendix A. We also updated Fig. 1 accordingly for clearer visualisation of the methodology. To avoid confusion, we have deleted the words 'multinary' and leave that detail to the reading of G21.

We have recalled Eq (8) in G21 in the Appendix A as Eq (A4) and briefly elaborate on that from lines 99 to 104 in the revised manuscript.

**RC1\_4:** Setting of the inverse problem. Eq 1 alone can be misleading. It is worth recalling that it comes from a linearization of the problem. I didn't find the information about the iterative scheme that is used to solve the non-linear problem, I guess it's a steepest descent.

**Reply:** Thank you very much for pointing this detail. We have added information about the iterative scheme in the revised manuscript between lines 101 and 103 With the related reference:

" The system of equations is then dumped into a least-squares system of equations that are solved using the least-square algorithm (Paige and Saunders, 1982). "

This information about the iterative scheme is added before Eq (1) to make it more comprehensible. We have also added a short note about this scheme in the introduction section in the revised version (line 59). Other than that, it has been stated in manuscript (lines 100 and 138) that the system of equations is being solved in least-square framework.

Eq (1) is further updated based on your suggestion in comment No. 9.

Below are some remarks about the text:

**RC1\_5:** 1) You use throughout the text the notion of "rock unit". It seems to me that

"Geological or geophysical unit" would more appropriate since you can deal also with sand, clay, salt, etc.

**Reply:** After re-examining the text of the original manuscript, we fully agree that using the terms 'geological units', 'lithological unit', 'geophysical units', and 'rock units' in different parts of the manuscript might be confusing. We appreciate you pointing this in your comment. In the revised manuscript we have replaced all other three terms with 'rock unit' as it has been widely used in geoscientific papers as referenced below: (Giraud et al., 2021b, 2021a; Kieu and Kepic, 2020; Witter et al., 2016, Astic et al., 2020; Lelièvre et al., 2010; Morris et al., 2007). In this study, the defined units are geologically plausible geophysical units so we have replaced all terms with 'rock unit' that covers both concepts.

**RC1\_6:** line 83: you introduce signed-distance values to interface calculated by FMM. Without further explanation, this sentence is totally incomprehensible. Outline is inappropriate, use boundary or interface instead.

**Reply:** This sentence has been modified along with modifying the methodology summary section based on your comment No. 3. We have provided an illustration of the signed-distance values to the interfaces in the appendix. We have also replaced the word 'outline' with 'boundary' in the revised manuscript. We'd like to thank you for your suggestion.

**RC1\_7:** 2) Line 85: the sentence where you transform a "signed distance" to a multinary structure (???) using a smeared-out Heaviside is obscure.

**Reply:** We agree that in the original manuscript there was obscurity in the definition of the signed-distance and Heaviside function. The mentioned sentence has now been transformed into clearer sentences along with equations and figures in the appendix.

**RC1\_8:** Line 91-102: This paragraph is very confusing and for me incorrect. The sentence "Initializing the model space..." is confusing. **m()** is the model function that links the modelled data to the parameters, through the signed distance  $\Phi_k$ . It is not a space, neither in a mathematical sense nor in geometrical sense. And you do not "initialize" a model, unless you talk about the initial (trial) model, you "define" it. You'd rather stick to G21 formulation in this part.

**Reply:** Thank you for this important remark. We agree that we have used incorrect terms in the manuscript and it was a mistake. In the revised text, we have removed the terms: 'space' and 'initialising' as suggested. We have used your suggested equation (in the following comment) for Eq (1). The entire paragraph and the equation have been modified.

**RC1\_9:** Eq 1 is totally confusing since it mixes a general and an iterative formulation. What is **d**calc? It is never defined. I suggest to rewrite this paragraph according to a more standard way of presenting inverse problems:

a) You are interested in solving a discrete inverse problem whose direct formulation is:

**d**=**g**(**m**); **d**= data; **m** parameter to be inverted; **g**() the direct function, non linear in our case.

b) You decide to solve this non linear problem using a gradient type method

base on a 1st order Taylor expansion

 $\mathbf{g}(\mathbf{m}) \approx \mathbf{g}(\mathbf{m}0) + \frac{\partial g}{\partial m}\Big|_{m_0} (m - m0)$

c) Considering the parameters of your direct problem:  $\mathbf{m}=\mathbf{m}(\Phi,\rho)$  in which density is kept constant, this turns into:

 $\mathbf{g}(\mathbf{m}) \approx \mathbf{g}(\Phi \mathbf{0}) + \frac{\partial g}{\partial \mathbf{m}} \frac{\partial m}{\partial \Phi} \Big|_{\Phi_0} (\Phi - \Phi_0) \leftrightarrow \mathbf{g}(\mathbf{m}(\Phi)) = \mathbf{g}(\Phi \mathbf{0}) + J^{\Phi} \delta \Phi$

d) And you decide to iteratively minimize in a least square sense:  $\Psi_{i+1}^r = \|\mathbf{d}^{obs} - g(\Phi_i) - J^{\Phi_i} \delta \Phi\|_2$

where now  $\mathbf{d}^{calc} = \mathbf{g}(\Phi_i)$  is defined as the result of the direct problem at iteration i. Please note that compared to your eq1. I have a sign difference. You never use the residuals r that is defined in line94, is it necessary?

**Reply:** Thanks for writing the step-by-step equations toward linearization of this inverse problem. We consider your commentary about the mathematical notation very appropriate and have accordingly corrected the Eq (1) based on your suggestion.  $\mathbf{d}^{calc}$  is now accordingly defined prior to Eq (1). Residuals (r) that encapsulate the difference between calculated and observed datasets are already used in equation 4 so we think it is necessary to keep it and also to be consistent with equations in G21.

**RC1 10: 2.2 Regularization level-set inversion**

Sentences in lines 118-121 are confusing and the statement is incorrect, this regularization does not "encourage the  $\delta \Phi$ update to reach specific values stored in **q**", but it does "encourage the product  $W\delta\Phi$ update to reach specific values stored in q" which is quite different (imagine that W is a Laplacian, or a smoothing operator). Since at this point neither **W** nor **q** are defined, it is difficult to understand what the authors mean.

I suggest that the authors replace the text that is too general by more precise details that are given later in the text. What is the exact size of **q** vector?

Do you try to impose something like  $\begin{vmatrix} W_S \delta \Phi \\ W_P \delta \Phi - v \end{vmatrix}$  minimum?

**Reply:** Thank you for the suggestion. We corrected the quoted sentence and replaced the general text with more precise details. W terms are defined from lines 127 to 130 in the revised text and also in section 2.3. In this case, W is not Laplacian or a smoothing operator but weight for regularization terms to encapsulate prior (or constraint in this study) information. We think that your suggested notation is very appropriate so we have deleted the notation **q** and, have taken the suggested formulation as a replacement for Eq (3) in the revised manuscript, as it is more comprehensible.

**RC1\_11:** Besides, why do you mix these two constraints simultaneously? What is the difference between imposing eq 4 rather than  $||W_S \delta \Phi|| + ||W_P \delta \Phi - v||$ minimum? or  $(\delta \Phi - \delta \Phi_{prior})^T C_{\Phi}^{-1} (\delta \Phi - \delta \Phi_{prior})$ ? Should we interpret Wp as a geometrical mask (rather than a weighting) that allows fixing some specific values of boundaries in the different geological units?

**Reply**: Thank you for your comment. We impose two constraints simultaneously since one of them is applied separately to each unit while the other is a global term. We have added a note on this in the revised manuscript at lines 140-141. The two equations are basically pointing at the same

concept, the only reason we wrote the formulation as Eq (4) is that  $W_S$  and  $\delta\Phi$  are vectors and the norm might not have a meaning for the product of two vectors.  $||W_S\delta\Phi|| + ||W_P\delta\Phi - v||$  can be pointing at the same as Eq (3) in the original manuscript if  $\delta\Phi$  is defined as the difference of  $\Phi$  function between two successive iterations ( $\Phi_{k+1} - \Phi_k$ ) and not as the difference between  $\Phi$  and  $\Phi_{prior}$ . However, as was suggested in the previous comment, we have changed the formulation to a closer notation to G21 for more consistency. Eq (3) is changed based on your suggestion and so Eq (4) is now omitted. We have also moved more precise details to the beginning of the paragraph to make the paragraph clearer as was suggested.

As for a reply to the second part of this comment, Wp is interpreted as a weighting term in which geometrical mask can be a particular case of these weights.

**RC1\_12: 2.3** The sentence on line 149 is incomprehensible, and the full paragraph from 148-153 confusing.

**Reply:** The entire paragraph is aiming at illustrating the addition of seismic information as a constraint to the inversion. It is using Figure 1 to illustrate. We have slightly updated Fig. 1 in the manuscript and so we updated the corresponding text in the mentioned paragraph for more clarity.

**RC1\_13: 2.4.** Your explanations are ok, however it is difficult to grasp the influence of this topological rule enforcement on your results. Could you comment on the effects of this processing on the synthetic case for instance?

**Reply:** Thank you for your comment. One example of the effects of applying this topological rule is shown in Fig. 1\_1 attached to this document. In this example which is the top view of the unconstrained case (Fig. 5d), it could be seen that the topological rule has reduced the nucleation (inclusion) of the background lithology (blue unit) into the yellow unit. We have added this Figure to the revised manuscript in Appendix B.

**RC1\_14: 3.** Figure 5 caption: what do you mean by difference between "new data and synthetic", what are the new data?

**Reply:** Thank you for noting this detail. It was a mistake to write 'new data' in the caption. What was meant by new data was, calculated datasets from the final inverted model. It has been corrected now.

**RC1\_15: 3.** A general question: in 4.3 and 4.4 you choose to build a starting model from the inversion of density only, then invert for the interfaces only in a second step. Why don't you try to invert simultaneously for interfaces and density values in the different units?

**Reply:** Doing the simultaneous inversion for density and interfaces is not a trivial task to do and is not the main focus of this study. Implementing such a technique in the presented level-set study can be a new area of active research that requires the reformulation of the inversion problem. We have added a couple of sentences about this fact to the discussion section from lines 516 to 518 as: "In this area of study, due to lack of availability

of petrophysical datasets we first implement physical properties inversion followed by a constrained level-set inversion. Although simultaneous inversion for density and interfaces would be beneficial to be done in this region, is not a trivial task to do and is beyond the scope of this study".

**RC1\_16:** Line 384: "due to … sections". Use a direct formulation instead: "We present … because…"

**Reply:** Thanks for the suggestion. The sentence was corrected in the revised text based on your comment.

**RC1\_17: 4.5.2** I do not understand which geometrical constraints you apply from the seismic profile. On figure 2d for the synthetic case, we clearly see that your constraint follows the geometry of the reflector. What about results obtained on figure 12?

There are no clear reflectors such as those of synthetic examples, but rather several general eastward dipping trends. Which constraints do you apply? Could you provide a plot of these constraints along the 2D section?

**Reply:** Thanks for noting this, we agree that visualising a constraint like Fig. 2d can be advantageous for the case-study. Not seeing a clear reflector in seismic images in a hard-rock environment is quite common. Also, even if the reflectors were detectible, for assigning different rock units to the reflectors, petrophysical constraint, and other integrated interpretations would be necessary to provide a clear interpretation of the seismic image as what has been shown in the example sections. For the case study section we have done some post-processing on the seismic section to extract some features from the most obvious reflectors so that they could be used for interpretation and extracting constraints. We used Energy envelop of the seismic traces (be it a function of amplitude) to enhance the effects of reflectors within their neighbourhood. Including such a section in the paper requires us to talk about the entire process that was implemented on the seismic section. We believe that people working on seismic datasets can find this way or other alternatives to extract the most detectible features from images easily. We have attached a Figure in this document regarding the section from the seismic image that was used for extracting constraints in the case study section (Fig. 1\_2). The original size of the seismic section was (4001\*4501) showing the high-resolution image, while for using it as a constraint as explained in the method and introduction sections, it should have the same size as the gravity grid section (20 \* 154). Therefore, the interpretations need to be up-scaled and projected onto the gravity inversion mesh. The resulted section is eventually used for further interpretation and constraint extraction. We have used the presented interpretation of seismic datasets in Lindsay et al (2020) to assigned reflectors to different rock units as shown in Fig. 1 2. We prefer not to include the attached section in the manuscript because 1) along with the image we should provide a long section about generating this image 2) the process is mostly related to post-processing technique and is only applicable if the datasets are noisy like what is used in this

study so it might reduce the consistency of the paper and will be beyond the scope of this paper.

We have now added a sentence stating that we use Lindsay interpretations to produce the constraints at lines 448-449.

**RC2\_18:** You mention on line 459 that Goleby et al. (2004) and Lindsay et al. (2020) use different seismic profiles. On line 341 you mention that you use Goleby interpretation. Is your seismic profile coming from the 2004 study or the 2020 study?

**Reply:** Thank you for this careful remark. After re-reading the text we agree that the original manuscript was confusing. We have made corrections to the text at lines 360-361 and 366-367 to be consistent and stating that we use interpretation from Lindsay et al. 2020. The seismic profile in both studies uses the same data, but Lindsay et al. 2020 reprocessed the data using updated techniques which resulted in slight differences in reflector positioning, signal-to-noise ratio, and detecting the dip of some events. Lindsay et al (2020) use the reprocessed data for interpretation, while Goleby uses the older version. We use the seismic line that has been reprocessed later and is utilised by Lindsay et al (2020). The presented result in Lindsay et al (2020) does not very much contradict with Goleby's result. The main difference between the two studies has been pointed at lines 479 to 481.

**Fig 1\_1:** Top view of the hard-rock synthetic example at the depth of 150 m. (a) and (b) show the results of the unconstrained inversion without applying the morphological closing to the model and after applying the morphological closing constraint respectively.

**Constraint Section**

**Fig 1\_2:** Up-scaled extracted seismic reflectors from the original seismic image for the constraining purpose of the level-set inversion in the Yamarna Terrane.

---

## Author Comment (AC2)

**Reply to Anonymous Referee # 2 comments**

Dear reviewer,

Thank you for the detailed and thoughtful comments. We took all your comments into consideration. We have copied and numbered your comments and addressed them point-by-point below.

Sincerely,
Mahtab Rashidifard, Jeremie Giraud, Mark Lindsay, Mark Jessell, Vitaliy Ogarko
* * *
**RC2_1:** This paper describes an application of a level set method to the problem of reconstruction density from gravity data in 3D with additional constraints obtained from 2D seismic survey. The constraints are implemented as a regularization penalizing the evolution of the level set functions where the model is deemed to reliably constrain by from prior information, in this case an inverted 2D seismic section. The method is applied in synthetic examples and to a field data set from Yilgarn Craton in Western Australia.

I think the main contribution of this paper is the adaptation of the methodology to the specifics of the area studied in the field data example and the specific kinds of data available there. I recommend accepting the paper with some revisions.

> **Reply:** We would like to commence by thanking the referee for providing a constructive and exact review of the manuscript, as shown by this precise summary.

**RC2_2:** Comments on the substance of the paper:

I think the theory section is adequate, considering that a previous paper by the authors where the methodology is explained in more details is referenced. Examples section is adequate in scope, however some results were a bit difficult to interpret. I believer that several figures might benefit from editing and some additional comments would help.

> **Reply**: Thank you for your feedback about the substance of the manuscript. We have made some modifications to the example section and made the results clearer based on your comments.

**RC2_3:** Specifically, I have the following suggestions:

**RC2_4:** Lines 106-108: The meaning of the sentence is hard to understand, I suggest reformulation for clarity. Gravity level set inversion actually is quite well-posed when gravity values are known and does not need much regularization, in my experience. When minimum length/area regularisation is used, usually it does not lead to overly simplified shapes when the data constrain the problem well.

**Reply:** Thanks for the hint. The mentioned sentence has now been reformulated in the text (line 117-119:) "Minimizing the length of the geometries, generates shapes with the smallest area and regularizing these inversion problems can be limited to the specific shape of units that can introduce a bias towards unrealistically simple geometries."

This is a fair point. In the referenced level-set inversion methods (Li et al. 2016, Li et al. 2017, Zheglova et al. 2018) using known constant physical property values has already reduced the non-uniqueness of the inversion problem however, non-uniqueness of problems are even reduced more using prior information in additional regularization terms. In the introduced approach we follow a different strategy for the application of regularizations. In this manuscript, we use area-specific regularizations, which require minimum updates of the model (not minimum structure as per the works referenced above). What we enforce as constraints in this study comes from a higher resolution method and it might, on occasions, violate the minimum area constraint in cases where seismic reflectors show a non-smooth surface while being perfectly valid geophysically.

**RC2_5:** Lines 125-126: It would be nice to reformulate the sentence, because its meaning is hard to understand.

**Reply:** The mentioned sentence is now re-written as (line 134-135:), "We show that by updating global and local regularization terms with low-uncertainty information from seismic datasets, they can act as global and local constraints."

**RC2_6:** Line 129: What exactly does "uniform" mean here?

**Reply:** Thank you for noting this detail. It was a mistake to use the term "uniform" in the text. The word is deleted in the revised manuscript.

**RC2_7:** Line 135: Could you please elaborate a little bit on the difference of the effect of Ws and Wp?

**Reply:** We have updated and modified the theory section based on Referee_1 comments to elaborate more on the effects of the constraints. We have emphasized that $W_s$ is a constraining term that includes all rock units at once while $W_p$ considers each rock unit separately. We have added this to section 2.3 based on your suggestion (from line 140 to 142:) "We use global regularization term to encapsulate the information about all lithologies in one vector while local terms are defined to include different lithologies separately in the inversion problem."

**RC2_8:** Lines 141-142: Could you please make it clearer, what is meant by "arrays", also what the "sample section" means here.

**Reply:** Thank you for pointing that we have not used understandable terms in this section. The mentioned sentence is explaining part of Fig. 1. In the revised manuscript we have updated Fig. 1 so we have updated the corresponding explanation in the text (from line 153- 154 :) "All parts of the 3D model that lie within the constraining section are weighted accordingly". In this figure, it

is assumed that within a 3D model, there is a section (colored section) that is going to be used for constraining. So in this sentence, the message is that parts of the 3D model that overlays the 2D section are given different weights for the constraining purpose.

**RC2_9:** Lines 148-150: The meaning of the sentence is not clear.

> **Reply:** The mentioned sentence is reformulated in the new text after updating Fig. 1 (line 161-162:), "As shown in Fig. 1b, in a 3D volume with the same size as the model, the extracted section along the seismic profile is weighted accordingly for lithology 2."

**RC2_10:** Line 165, Figure 1: The top two images on this figure need labels, captions and more explanation. How do the bottom figures, especially figure 1b fit into the right top image? Why do images (a) and (b) show two different plots of the same matrix Ws2?

> **Reply:** We agree that Fig. 1 needs modification. This figure has been updated and the corresponding explanations have been reformulated in a clearer fashion in the revised manuscript. In the manuscript in the method section (lines 166 -171 of the revised manuscript) it has been already explained that why two kinds of $W_{s2}$ can be defined. This should be clearer in the revised text given that Fig. 1 has been updated.

**RC2_11:** Section 2.4. As far as I understand, this regularization prevents small pieces of one lithology to be isolated inside another lithology, reducing fragmentation of the model, but it's not quite clear why such a constraint needs to be applied. What is meant by "nucleation"?

> **Reply:** Thank you for the hint. We have updated the sentence (at line 201-204:) "We take advantage of a certain type of the morphological rules of image processing techniques to prevent the nucleation of a given unit into another and for the model to obey topological rules. This becomes important for retaining the integrity of the predefined unit boundaries during the inversion and ensuring geological plausibility of the inverted model (age and deformation history)"
> The exact meaning of Nucleation in geophysical inversion is the inclusion of one lithology into the other. We have added Fig. B1 to Appendix B in the revised manuscript to show the effect of applying this constraint on the second example.

**RC2_12:** Line 218: I wouldn't call the starting disc model random. Maybe it is better to use a different word to describe the choice of the initial model.

> **Reply:** Thank you for noticing this detail. The starting model is not random and we corrected the term. We replaced the word "random" with "simplistic" as in line 233.

**RC2_13:** Lines 254-255: The sentence seems to contradict later sentences: it says that the seismic section is only applied in the construction of the initial model. However, around line 260 it is said that the reflectivity from the seismic section is also used as a constraint during inversion.

**Reply:** After re-examining the text we agree that the sentence is misleading. We have reformulated the sentence to prevent further misunderstanding. What is meant by mentioned sentences is: for generating the starting model, we have assumed that only the seismic section is available (meaning that the starting model follows the seismic section only and not the gravity datasets). The sentence is replaced (at line 270:) "…we generate the starting model using only information from the seismic section".

**RC2_14:** Line 286: "The difference between the structural similarities" and "an indication of the applicability of the approach to spatially distributed constraints in the level-set inversion" -- these two phrases are hard to take in and could be simplified for clarity.

**Reply:** Thanks for the suggestion. We simplified the sentence as (line 303-304:) "This implies that the method can be applied to real case scenarios where gravity and seismic datasets with different coverages are available."

**RC2_15:** Line 295, Figure 5: It might be nice to show the true and inverted models from the same angle.

**Reply:** Thanks for the suggestion which we agree with. We have changed the view of Fig. 4a to the same slices and same angle as Fig. 5 which represent the true and inverted model respectively. Now, Fig. 4 and Fig. 5 are in the same direction for viewing and the same slices are being used.

**RC2_16:** Line 440, Figure 12: Compared to the starting model, Figure 10b, it appears that the green, blue and brown units have switched placed and moved away from their original locations quite a lot. It is usually hard to recover the shape of a unit if there isn't some overlap between the initial and true unit location, so such a result doesn't look plausible. Considering that also the evolution of the level set function was suppressed at the seismic section, this final reconfiguration of the facies is very unlikely.

**Reply:** Thank you very much for noting this detail. As you mentioned, there was a mistake in colorbars of Fig. 10 and 15 and also some mistakes in density contrast values in the table which had led to different colorbars. We have updated the density contrasts in Table 1 and color-map of the Figure 10 and plotted different colorbar for (a) and (b) so that the color of the units matches with Figure 12. Having the color bar corrected, the problem about switching units is resolved now.

**RC2_17:** Are you sure that the units are plotted in the correct color? The color bar from Figure 15 would make the models on Figure 12 much more plausible and consistent with Figure 10 and the discussion. This needs to be fixed or explained. It would be helpful if Figures 10, 12 and 15 used the same color scheme, so I suggest replotting Figures 12 and 15 using the color scheme of Figure 10.

**Reply:** This is a very correct point which we are really appreciated for pointing it. As was suggested from the previous comment, we updated the color-scheme of Figures 10 and 15 to be compatible with Fig 12. Fig 10 is replotted with new colorbar and for Fig 15 we have updated the seismic section background color

to be more comparable with Fig 10 and 12. The seismic section in Fig. 15 now is in grayscale so when overlain with the model, slight changes in colors are inevitable.

**RC2_18:** Also, it would help to plot the final models on the same set of axes as the initial model in Figure 10a, to better visualize the shape changes of the bodies. It would also help to plot the initial and both final models along the seismic section overlain on the seismic image as in Figure 9d.

> **Reply:** We agree that plotting the results in 3D might be a good visualisation. However, plotting all of the units in one frame in 3D for this case-study section will be a bit messy and we believe that the final model if plotted in the same way as Figure 10, won't be informative enough for the conclusion. The main focus is to compare the resulted constrained model along the seismic section, which we have included in the manuscript as Fig. 15. For showing that the resulted 3D model can be messy we have provided 2 figures (Fig. 2_1 and Fig. 2_2) that show 3D visualization of the Yamarna Greenstone belt as a sample in this document.

**RC2_19:** Lines 443-444: Again, it's a bit difficult to compare the initial and final models and also see how well the final models fit the constraints. If Figures 12 (a) and (b) were plotted on the same kind of axes as Figure 10, perhaps this would make understanding the changes in the model after the inversions easier.

> **Reply:** In the original manuscript, there were some mistakes and contradictions between colorbars. We have corrected Fig. 10 colorbars based on previous comments. Now the two images can be compared easier given that the colorbars are fixed.

**RC2_20:** Lines 448-449: The fact that the models differ in constrained parts and do not much differ elsewhere seems to indicate that the information in the constraints does not quite agree with the information in the gravity data. I wonder if it might indicate that the constrained inversion result is incorrect elsewhere, or that the constraints themselves are incorrect? Could the authors comment on this? Have you tried a synthetic, in which the constraints were assumed correct, while they weren't, to see how robust the inversion is to such errors? There is a relevant comment on uncertainty in the conclusion section, but it is a bit far below and hard to tie to this particular place, so it would be nice to make a comment here.

> **Reply:** By definition, the seismic section is uncertain and the model proposed is one among the possible ones that gravitate around the causative model. As a consequence, such seismic constraints might stir the gravity in the right direction, and the information from the gravity data is used to adjust that model, not completely wildly but somehow in accordance with the seismic. On the other hand, this area as explained in the manuscript is poorly known and unfortunately, the petrophysical constraints are not available in the studied area. The seismic section also being in a crooked 2D line and with a low signal-to-noise ratio (hard-rock) results in high imaging and interpretation uncertainties.
> About the second part of the comment, yes it has been tested and can be well referenced later in the revised manuscript once the abstract is online. We have

provided this experiment in a conference abstract to AEGC which showed that in case of the wrong constraint the inverted model is still plausible compared to the true model. Also, the synthetic case of the SEG salt dome shows an example where the constraining information is not complete.

**RC2_21:** Line 465, Figure 15: The numbers and colors on the color bar are out of order and inconsistent with Figures 10 and 12. Could this figure be replotted in the color scheme of Figure 10, for easier understanding? How does the unconstrained inverted model compare with the prior interpretations? Could you perhaps show an example?

> **Reply:** Thank you very much for the suggestion, we have re-plotted Figure 15 with the grayscale seismic image that has the least effect on the color scheme. We have corrected and adjusted the colorbar based on consistency with figures 10 and 12 too.
>
> Also, we have added a new figure (Fig. 2_3) in this document that shows the suggested comparison along the seismic line. We think that this figure is not necessary to be included in the manuscript as a comparison between the constrained inversions overlain with existing seismic interpretation is already provided in Figure 15. The shape of the recovered models in Fig. 2_3a and Fig. 2_3b can be seen in Fig 10b and Fig. 12a in the manuscript and the overlain with seismic image might not be informative for the conclusion.

**RC2_22:** Line 505: Could Yamarna Greenstone unit be marked on Figure 12b? It would probably help to better appreciate the shape changes, if this body were plotted in 3D.

> **Reply**: It could be a great idea to plot the Yamarna unit in 3D if it wasn't messy as what is shown in Fig. 2_1 and Fig. 2_2. We think is messy and doesn't add much information to the manuscript.

**RC2_23:** Line 509-510. Again, this is a bit difficult to see from the plots on Figure 12, I think a 3D plot of this body would help.

> **Reply:** We have addressed this issue by replying to comments No. 18, 19, and 22.

**RC2_24:** Technical comments:

Lines 111-112: "in the same fashion as that smallness terms regularize inversion problems (Calvetti et al., 2000)", remove "that".

Line 118: I suggest using small "w" if the sentence is continued, or start a new sentence with "Here" with the capital "H".

Line 145: "as follow:" -> "as follows:"

Lines 161-162: Word "plausible" seems to be out of place here. It's not quite clear what this sentence grammatically means.

Line 165, Figure 1: In the caption, second line remove "of" from "Distribution of constraint matrix of from lithology 2".

Line 256: Replace "area. In" by "area, in", otherwise the second sentence is grammatically incomplete.

Lines 480-482: The first sentence grammatically needs improvement. The next small sentence needs to be reformulated for style.

**Reply:** Thanks for all of the technical comments which we found really useful to improve the text passage. The technical comments were all applied and the text was modified in the revised manuscript.

[Figure]

Fig 2_1: 3D visualisation of Yamarna Greenstone belt within the unconstrained inverted model

[Figure]

Fig 2_2: 3D visualisation of Yamarna Greenstone belt within the constrained inverted model

[Figure]

Fig 2_3: Comparison of the models (starting, unconstrained, constrained) overlain with the seismic image.